# Bandits with many optimal arms

**Rianne de Heide**
INRIA Lille and CWI Amsterdam
r.de.heide@cwi.nl

**James Cheshire**
Otto von Guericke University Magdeburg
james.cheshire@ovgu.de

**Pierre Ménard**
Otto von Guericke University Magdeburg
pierre.menard@ovgu.de

**Alexandra Carpentier**
University of Potsdam
carpentier@uni-potsdam.de

## Abstract

We consider a stochastic bandit problem with a possibly infinite number of arms. We write $p^\star$ for the proportion of optimal arms and $\Delta$ for the minimal mean-gap between optimal and sub-optimal arms. We characterize the optimal learning rates both in the cumulative regret setting, and in the best-arm identification setting in terms of the problem parameters $T$ (the budget), $p^\star$ and $\Delta$. For the objective of minimizing the cumulative regret, we provide a lower bound of order $\Omega(\log(T)/(p^\star\Delta))$ and a UCB-style algorithm with matching upper bound up to a factor of $\log(1/\Delta)$. Our algorithm needs $p^\star$ to calibrate its parameters, and we prove that this knowledge is necessary, since adapting to $p^\star$ in this setting is impossible. For best-arm identification we also provide a lower bound of order $\Omega(\exp(-cT\Delta^2 p^\star))$ on the probability of outputting a sub-optimal arm where $c > 0$ is an absolute constant. We also provide an elimination algorithm with an upper bound matching the lower bound up to a factor of order $\log(T)$ in the exponential, and that does not need $p^\star$ or $\Delta$ as parameter. Our results apply directly to the three related problems of competing against the $j$-th best arm, identifying an $\varepsilon$ good arm, and finding an arm with mean larger than a quantile of a known order.

## 1 Introduction

In the classical stochastic multi-armed bandit model – see [37] for a recent survey – a learner interacts with an environment in several rounds. At each round, the learner chooses an *arm* to play, and receives a random reward from the associated probability distribution. Popular settings are respectively the fixed budget *cumulative regret setting* [41], and *best-arm identification setting* [18, 7, 1]. In the first setting, the learner is interested in maximizing the sum of rewards gathered – or minimizing the cumulative regret – and in the best-arm identification setting, the learner is asked at the end of the game to output a guess for the arm with the largest mean reward, and is interested in the quality of this guess – typically measured by the probability of error in the guess.

In most of the papers that concern this topic, it is assumed (i) that there is a single optimal arm, i.e. arm with highest mean, and (ii) that the number of arms is bounded and small when compared to the time horizon, i.e. the number of rounds where the player is allowed to choose an arm. However in many realistic applications, it is not the case, for example in image classification, mining of resources, personalized medicine, or hyperparameter tuning (see [5] for more examples). And while it is clear that in all generality, the task of the learner becomes unsolvable if the number of arms is too large, it intuitively makes sense that if the proportion of optimal arms is also large, this should help the learner.

35th Conference on Neural Information Processing Systems (NeurIPS 2021).

In this paper, we lift both assumptions summarised in (i) and (ii) and study both the cumulative regret and best-arm identification setting. See Section 1.3 for literature related to this that we will discuss later. We will focus on the *problem dependent setting* and will aim at characterising optimal learning rates depending on the proportion of optimal arms, and on the minimal gap between the mean of an optimal arm and the mean of a sub-optimal arm.

## 1.1 Setting

We consider a setting with a (potentially infinite) set of arms $\mathcal{A}$, which we call the *reservoir*. Each arm $a \in \mathcal{A}$ is associated with a probability distribution $\nu_a$, which we assume to be supported on $[0,1]$, and we denote its mean by $\mu_a$. Write $\mu^* = \max_{a \in \mathcal{A}} \mu_a$ for the highest mean[1], $\mu_{sub} = \sup_{a \in \mathcal{A}: \mu_a \neq \mu^*} \mu_a$ for the second highest mean, and $\Delta = \mu^* - \mu_{sub}$ for the associated minimal gap. We will focus throughout this paper on the case where $\Delta > 0$.

We further assume that there exists a partition $\mathcal{A} = \mathcal{A}^* \cup \mathcal{A}_{sub}$ such that each arm $a \in \mathcal{A}^*$ is optimal, i.e. $\mu_a = \mu^*$, and each arm $a \in \mathcal{A}_{sub}$ is sub-optimal, i.e. $\mu_a \leq \mu_{sub}$. We assume that the agent can pick arms uniformly at random from the reservoir $\mathcal{A}$[2], and this arm belongs either to the set $\mathcal{A}^*$ with probability $p^\star$, i.e. there is a proportion $p^\star$ of optimal arms in the reservoir; or it belongs to the set $\mathcal{A}_{sub}$ with probability $1 - p^\star$, i.e. there is a proportion $1 - p^\star$ of sub-optimal arms in the reservoir.

The learner interacts with the environment in several rounds $t = 1, 2, \ldots, T$, where we fix the time horizon $T$. At each round $t \leq T$, the learner chooses an arm $a_t$ by either picking a new arm from the reservoir $\mathcal{A}$ or playing a past arm, and gets a reward $Y_t \sim \nu_{a(t)}$. The arm choice depends only on the past observations, the past arm choices, and possibly some exogenous randomness. The rewards for each arm $a$ are i.i.d. random variables with mean $\mu_a$ unknown to the learner.

**Cumulative regret setting.** The first setting we study is that of minimizing the *cumulative regret*. This setting enforces the *exploration-exploitation trade-off*: the learner needs to balance exploratory actions to get a better estimate of the reward distributions, and exploitative actions to maximize the total return – and minimise the associated cumulative regret. The cumulative regret is the difference between the sum of expected rewards the learner would have obtained by only choosing the arm with the highest mean reward, and the sum of expected rewards she actually collected:

$$R(T) = \sum_{t=1}^{T} \mu^\star - \mu_{a(t)}.$$

**Best-arm identification setting** In the second setting we study, we are interested in identifying an arm with the highest mean reward. At the end of $T$ rounds, the agents selects an arm $\hat{a}_T$ and aims at minimising the probability of outputting an arm with sub-optimal mean:

$$\mathrm{e}(T) = \mathbb{P}(\hat{a}_T \notin \mathcal{A}^*).$$

A closely related popular measure of error is the *simple regret*, which is not discussed in this paper.

**Equivalent settings** Firstly, our setting is directly applicable to the problem of competing against the $j$-th best arm, where we assume w.l.o.g. the arms to be ordered according to their means. Indeed our setting translates to this if we replace $p^\star$ by $j/K$ and $\Delta$ by the gap between the $j/2$-th and the $j+1$-th best arm, i.e. $\Delta = |\mu_{j/2} - \mu_{j+1}|$. Secondly, our setting is directly applicable to that of identifying an $\varepsilon$ good arm, and thirdly, our setting is directly applicable to finding any arm in the reservoir with a mean larger than the quantile of a known order – see the discussion in Section 1.3.

## 1.2 Contributions

We characterise the optimal learning rates both for the cumulative regret setting, and for best-arm identification, for our problem described above. We characterise the optimal learning rates in terms of the problem parameters $T, p^\star$, and $\Delta$.

In order to describe our results, let us write for $\bar{\Delta} > 0$, $\bar{p}^\star \in [0,1)$: $\mathfrak{B}_{\bar{\Delta}, \bar{p}^\star}$, for the set of bandit problems whose reservoir distribution is such that $p^\star \geq \bar{p}^\star$ and such that $|\bar{\mu}^* - \mu_{sub}| \geq \bar{\Delta}$.

---

[1]We assume that it is attained for some arm(s).

[2]In case of infinite $\mathcal{A}$, one can obviously not sample from a uniform distribution. Our analysis extends to general distributions on $\mathcal{A}$.

**Cumulative regret**  We provide an algorithm, *that takes $p^\star$ as a parameter*, that is such that (see Theorem 1)

$$\mathbb{E}R(T) \leq O\left(\frac{\log T \log(1/\Delta)}{p^\star \Delta}\right).$$

Conversely, we prove in Theorem 2 that for $\bar{p}^\star \leq 1/4$ and $\bar{\Delta} \leq 1/4$, and for any algorithm, there exists a problem in $\mathfrak{B}_{\bar{\Delta}, \bar{p}^\star}$ such that

$$\mathbb{E}R(T) \geq \Omega\left(\frac{\log T}{\bar{p}^\star \bar{\Delta}}\right).$$

These two bounds match up to a multiplicative factor of order $\log(1/\Delta)$. They highlight the intuitive fact that we should pay the number of arms in the rate only relative to the number of optimal arms – i.e. only through $p^\star$. Indeed, the probability of picking an optimal arm in the reservoir when sampling uniformly at random being $p^\star$, if we sample about $1/p^\star$ arms at random from the reservoir, we will have sampled one optimal arm with constant probability – so that $1/p^\star$ plays the same role as the number of arms.

Having said that, there is a main conceptual difficulty in order to get a rate that is tight in terms of its dependence in $T$. If we sample only $1/p^\star$ arms from the reservoir, the probability of having no optimal arms in the chosen set of arms is also a constant – so that the regret is linear in $T$. It is therefore essential to sample *more* arms. In order to have a logarithmic regret in $T$, we need to sample at least about $\log T/p^\star$ arms from the reservoir – in which case at least one of them is optimal with probability polynomially decaying with $T$. But if we do this, we get a regret of order $\frac{(\log T)^2}{p^\star \Delta}$, as there are about $\log T/p^\star$ sub-optimal arms whenever $p^\star$ is not too close to 1. This is much larger than the bound that we have, where the dependence on $T$ is only $\log T$. In order to achieve this bound, we need to take into account the fact that when sampling $\log T/p^\star$ arms from the reservoir, there is typically not just 1, but $\log T$ optimal arms with high probability – and leverage this fact both in our algorithm and in the associated proof. We describe this in more detail in Section 2.1.

**Best-arm identification**  We provide an algorithm *that does not take $p^\star$ as a parameter*, such that,

$$\mathrm{e}(T) \leq O\left(\log(T) \exp\left(-c\frac{T\Delta^2 p^\star}{\log(T)}\right)\right),$$

where $c$ is some universal constant. Conversely, we prove that for $p^\star \leq 1/4$ and $\Delta \leq 1/4$, and for any algorithm, there exists a problem in $\mathfrak{B}_{\bar{\Delta}, \bar{p}^\star}$ such that $\mathrm{e}(T) \geq \Omega\left(\exp\left(-cT\Delta^2 p^\star\right)\right)$, where $c > 0$ is some universal constant. These two bounds match in order up to a factor of order $\log(T)$ in the exponential, it is an open question here whether this term is necessary or not.

These bounds highlight the intuitive fact that we should pay the number of arms in the rate only relative to the number of optimal arms – i.e. only through $p^\star$. As in the cumulative regret setting, if we sample about $1/p^\star$ arms at random from the reservoir, we will have sampled one optimal arm with constant probability – so that $1/p^\star$ plays the same role as the number of arms.

As in the cumulative regret setting, there is again a main conceptual difficulty in order to get a rate that is tight in terms of its dependence in $T$. If we sample only $1/p^\star$ arms from the reservoir, the probability of having no optimal arms in the chosen arms is also a constant – which is way smaller than the targeted best-arm identification probability. In order to have at least one optimal arm in the set of arms picked from the reservoir with a probability that decays exponentially with $p^\star T\Delta^2$, the number of arms that have to be sampled should be larger than $T\Delta^2$. But if we do this, we get an upper bound on the probability of error that is of constant order – which is much larger than the bound that we have. In order to obtain our upper bound, we need to take into account the fact that when sampling $T\Delta^2$ arms from the reservoir, there is typically not just 1, but $p^\star T\Delta^2$ optimal arms with high probability – and leverage this fact both in our algorithm and in the associated proof. We describe this in more detail in Section 3.1.

**Adaptation to $p^\star$: diverging pictures for cumulative regret and best-arm identification**  The algorithm for cumulative regret takes (a lower bound on) $p^\star$ as parameter, but the algorithm for best-arm identification does not take anything related to $p^\star$ or $\Delta$ as a parameter. And so, while our algorithm for best-arm identification is adaptive to $p^\star$ and $\Delta$, our cumulative regret algorithm is

adaptive to $\Delta$ but not $p^\star$. In Section 2.3 we prove that it is not just a weakness of our analysis, but that it is *impossible to adapt to $p^\star$ when it comes to the cumulative regret*. The phenomenon of adaptation to the problem hyper-parameters being possible for best-arm identification but not for cumulative regret, was observed earlier: In the $\mathcal{X}$-armed bandit setting [40] show it is impossible to adapt to smoothness and [24] further classifies the cost of adaptation in this case. [44] explore the cost of adaptation to $p^\star$ for the problem independent case where the number of arms is large.

## 1.3 Related work

**Finite and small number of arms.** The regret-minimization setting, introduced by [41], has been well-studied for *finite*-armed bandit models. Algorithms for this problem fall into several categories: algorithms based on upper-confidence bounds (UCB) for the unknown arm means [30, 3, 2, 11], algorithms that exploit a posterior distribution on the means, such as Thompson Sampling [42, 34], and many more such as explore-then-commit [20] and phased-elimination [19]. Logarithmic instance-dependent lower bounds have already been obtained in the seminal paper by [36], and were generalized later, e.g. by [10], see [21] for an overview and simple proofs. In the setting where the number of arms $|\mathcal{A}|$ is finite and not too large – much smaller than $T$ – a classical problem dependent upper bound on the expected cumulative regret is[3]

$$\sum_{a \in \mathcal{A} \setminus \mathcal{A}^*} \left( \frac{8 \log T}{\mu^* - \mu_a} + 2 \right) \le |\mathcal{A}_{sub}| \frac{\log T}{\Delta} + 2|\mathcal{A}_{sub}|. \tag{1}$$

The bound in the RHS is tight if all sub-optimal arms have the same gap $\Delta$. Moreover, this regret bound asymptotically matches the lower bound by [10] up to a multiplicative constant. In the case where there are infinitely many sub-optimal arms, on the other hand, this upper bound is infinite, *even when the proportion of optimal arms $p^\star$ is large and where one would hope for better performances*.

The fixed-budget best-arm identification setting was introduced by [7, 1] and has been widely studied. It is well-known that algorithms that are optimal for cumulative-regret minimization cannot yield optimal performance for best-arm identification [8, 33]. Write[3] $H = \sum_{a \in \mathcal{A} \setminus \mathcal{A}^*} \frac{1}{(\mu^* - \mu_a)^2} \le \frac{|\mathcal{A}_{sub}|}{\Delta^2}$. The bound in the RHS is tight if all sub-optimal arms have gap $\Delta$. It is proven by [1] that given $H$, there exists an algorithm such that the probability of misidentifying an optimal arm is of order $\exp(-cT/H)$, where $c > 0$ is some universal constant. In the case where there is *a single optimal arm* this bound is provably optimal [12] when $H$ is known. However, in the case where there are infinitely many sub-optimal arms this upper bound is larger than $1$ and thus vacuous, *even when the proportion of optimal arms $p^\star$ is large and where one would hope for better performances*.

Importantly, our results in both settings extend to finite bandits. Furthermore we do not need infinite $\mathcal{A}$ for our results to be near optimal. In the finite setting with $K$ arms and $p^\star K$ optimal arms the problem is strictly harder than one with $\frac{1}{p^\star}$ arms and a single optimal arm. Indeed, the latter problem would correspond to one where the learner receives, as additional information, a partition of the set of $K$ arms in $\frac{1}{p^\star}$ groups, where one of the groups contains all optimal arms, and the others are only composed of sub-optimal arms. One can then see that we match the classical UB and LB for the finite bandit problem, up to $\log(1/\Delta)$ terms.

**Large to infinite number of arms.** The setting with an infinite number of arms – and sometimes also many optimal arms – has been studied in different settings.

A setting that is very related to ours is the infinitely many-armed setting where a distribution is assumed on the reservoir – called the reservoir distribution. At each round, the learner can pull a previously queried arm, or a new arm that is sampled according to the reservoir distribution. A classical assumption on the reservoir is that the proportion of $\bar{\Delta}$-near optimal arms is of larger order than $\bar{\Delta}^{-\alpha}$ for any $\bar{\Delta}$. This setting been studied for both cumulative regret minimization [5, 43, 6, 17] and for best-arm identification [13, 4, 15]. A classical strategy is to select a subset of arms from the reservoir, large enough so that it contains a near optimal arm with high probability, and to use classical bandit strategies on these arms. The minimax order of magnitude of the cumulative regret is then $\sqrt{T} \vee T^{\alpha/(\alpha+1)}$ and for the simple regret it is $T^{-1/2} \vee T^{-1/\alpha}$.

---

[3] In the case where $\mathcal{A}$ is finite otherwise the quantity below is infinite.

Related results have also be obtained in the setting where the number of arms is finite, but large – i.e. $K > T$ – and under related assumptions on the frequency of near-optimal arms [44]. While our setting is extremely related to this setting, the assumption about the frequency of near-optimal arms differs in the above literature from the assumption we make in this paper. Their bounds are not dependent upon $\Delta$ – they assume $\forall k \in [K], \mu_k \in [0, 1]$, and instead focus on achieving semi adaptivity in regards to an unknown $\alpha^*$, where $\alpha^* := \inf\{\alpha : K/|S_*| < T^\alpha\}$. In the context of our setting $T^\alpha$ would act as a upper bound on $1/p^\star$. They propose an algorithm with user defined parameter $\beta$ that has no guarantees on regret for $\beta < \alpha$. And while our assumption is more restrictive, we also expect to obtain much smaller optimal rates. Our results differ from this stream of literature in the same way that, in the classical MAB, *problem dependent results differ from problem independent results.*

Another setting takes a regularity assumption on the reservoir distribution around $\mu^*$ – that is, the proportion of arms in the reservoir whose gap is of order greater than $\bar{\Delta}$ is bounded above by a function of $\bar{\Delta}$, typically $\bar{\Delta}^\alpha$, where $\alpha$ is the regularity coefficient. For best-arm identification adaptivity is possible without knowledge of $\alpha$ and [13] provide algorithms for the simple regret with LB matching up to $\log(T)$ terms. In the case of cumulative regret [43] and [6] again provide near optimal results but in the case of *known* $\alpha$. While the above literature considers a weaker assumption on the reservoir distribution, their results are also considerably weaker than our own. For best-arm identification they identify a sub optimal arm whose distance to the optimal arm is bounded polynomially with $T$. For cumulative regret the regret is bounded polynomially with $T$. These bounds are in both cases much larger than our bounds – which essentially reflects that their assumption are weaker.

Closer to our setting are the works [15] and [4], where they try to find any arm in the reservoir with a mean larger than the quantile of a known order (with respect to the reservoir distribution) with high probability. This can be seen as the fixed confidence version of our setting for best-arm identification where the order of the quantiles is our known proportion of optimal arms $p^\star$ and the gap $\Delta$ is the difference between the first and the second quantile of order $p^\star$. Precisely, [4] provide an algorithm that can find an arm above the quantile of order $p^\star$ with probability at least $1 - \delta$ in less than $H_{\Delta,p^\star} \log(1/\delta)^2$ samples on average, where $H_{\Delta,p^\star} \approx 1/(p^\star\Delta^2)$ is the problem dependent constant. The fixed confidence result of [4] translates, in the fixed budget setting, into an upper bound on the probability of error $e(T)$ of order $\exp(-c\sqrt{Tp^\star\Delta^2})$ where $c > 0$ is some universal constant – which is much larger than our bound for large $T$. Similarly, [16] consider the regret with respect to a fixed quantile of order $p^\star$ of the distribution of the means in the reservoir which is again quite related to the regret in our setting. They obtain an algorithm with a bound on cumulative regret of order $R(T) \leq O\big(1/p^\star + \sqrt{(T/p^\star)\log(p^\star T)}\big)$, for any $\Delta > 0$ – in this sense, this analysis is problem independent.

Also closely related is the paper [32] which deals with identifying an $\varepsilon$ good arm – in the case where there are many such $\varepsilon$ good arms, with high probability. Again this can be seen as a fixed confidence version of our setting, with the proportion of $\varepsilon$ good arms being equivalent to our $p^\star$. However, the focus of their results differs considerably to our own. Specifically, in our setting, Theorem 2 of [32] provides an upper bound on the expectation of a stopping time for epsilon good arm identification, of the order $\bar{\mathcal{H}} \log(\bar{\mathcal{H}})$ where $\bar{\mathcal{H}} \approx 1/(p^\star\Delta^2)\log(1/\delta)$ but this bound does not hold in high probability, which would be necessary if one wished to directly compare their results to ours. Indeed for the stopping time of their algorithm to be bounded in high probability one would need to pay a $\log(1/\delta)^2$ term, corresponding to $\exp(-\sqrt{\Delta^2 p^* T})$ in our setting, see Remark 4 in [32] and page 15 in the appendix of the full version [31]. The focus of [32] is instead to get more complete gap dependent bounds, considering also the gaps within the epsilon good arms but as mentioned their results cannot be applied directly to our setting and, as they point out, extending their approach to include high probability guarantees would be strictly sub optimal compared to our results.

We can also view the *most-biased coin problem* studied by [14] and [27] as a particular instance of our setting where all optimal arms are distributed according to a Bernoulli distribution $\mathcal{B}er(\mu^\star)$ and any sub-optimal arm is distributed according to the *same* Bernoulli distribution $\mathcal{B}er(\mu^-)$. The goal is then to identify an optimal arm with high probability with as few samples as possible. Precisely, [27] prove that they can find an optimal arm with probability at least $1 - \delta$ with $\log\big(1/(p^\star\Delta^2)\big)\frac{\log(1/\delta)}{p^\star\Delta^2}$ samples in expectation when $\mu^\star, \mu^-$ and $p^\star$ are unknown to the agent and with $\frac{\log(1/\delta)}{p^\star\Delta^2}$ samples if $p^\star$ is known. It is also worth mentioning the problem of $p^\star$ estimation for the biased coin problem.

For unknown $p^\star$ and $\Delta$, [39] describe, in the fixed confidence setting, the optimal learning rate for estimating $p^\star$, up to an additive error $\varepsilon$, of the order $\frac{p^\star}{\varepsilon^2 \Delta^2} \log(1/\delta)$.

The translation of the result from [27] to the fixed budget setting is much closer to our result, as it would provide a bound of order $\exp\left(-cTp^\star\Delta^2 / \log(1/(p^\star\Delta^2))\right)$ where $c > 0$ is some universal constant. This is very similar to our bound, but there is a main difference: we do not assume that there are just two possible distribution for the arms as [27] – the set $\mathcal{A}_{sub}$ of sub-optimal arms might contain arms of diverse means, all being at a gap more than $\Delta$ from $\mu^*$. This makes the problem *significantly more difficult* – in particular regarding the adaptation to $p^\star$ – since in our setting, it is impossible to estimate the minimal gap $\Delta$, see Section 5. In fact, extending to a more general reservoir is an open question of interest left at the end of the above paper.

Otherwise, there are some other formulations of the infinitely-many armed bandit problem that are quite popular, but very different from our setting, and that we mention here for completeness. Many works are devoted to the setting where there is some topological relation between the index of the arms, and the mean of the arms [35, 9, 23]. This setting is often referred to as the $\mathcal{X}-$armed bandit setting, and not related to our work as we do not make such topological assumptions. Finally, a paper in which the setting is close to ours, but where the goal is very different, is the one by [28]. The authors consider a partition of the (infinite) space $\Omega$ of K-armed bandit models $\nu = (\nu_1, \ldots, \nu_K)$, and want to identify for a given bandit model $\mu \in \Omega$ the correct partition component it belongs to.

**Fixed confidence to fixed budget setting**    In the fixed confidence setting for best-arm identification, given some $\delta > 0$, one aims to bound the expected number of samples one needs to correctly identify an optimal arm with probability greater than $1 - \delta$. With our best-arm identification upper bound (Theorem 4) in mind, we can essentially translate our result to the fixed confidence setting by considering $\delta = \exp\left(-\frac{Tp^\star\Delta^2}{\log(1/\Delta)}\right)$, and solving for $T$. This leads to a upper bound on the number of samples `Elimination` needs to be $\delta$-approximately correct of: $\frac{\log\left(\frac{1}{\delta}\right)\log\left(\frac{1}{\Delta}\right)}{p^\star\Delta^2}$. The papers [27] and [4] both deal with settings very related to our own but from the fixed confidence perspective. [4] deals with quantile estimation and as highlighted above their results can be applied to our setting but with a significantly worse bound on probability of error of order $\exp(\sqrt{Tp^\star\Delta})$. In [27] the problem of best-arm identification is tackled directly but with strong restriction on the reservoir distribution, they consider the case were all sub optimal arms are identically distributed.

**Pair matching**    An additional setting that can be seen in the context of our problem is that of pair matching. Here the learner is presented with a finite graph of nodes, $N$. The set of nodes $N$ is partitioned into 2 or more communities. The general idea is that nodes in the same community are more likely to be connected by an edge than those in separate communities. A simple and well studied situation is where the graph is generated according to a stochastic block model (SBM), see [25]. In this setting the probability of an edge forming between two nodes of the same community is $p$ and the probability of an edge forming between two nodes of differing communities is $q$, with $p > q$. Much of the literature is then concerned with identifying communities given complete access to the graph, see [38] and references therein. Of more relation to our specific setting is the paper [22]. Here the learner does not immediately observe the complete graph but is instead able to sequentially query whether two nodes are connected up to a budget $T$. Their objective is then to minimise their sampling regret, the number of times they query and edge between 2 nodes of differing communities. The problem can be viewed as a bandit problem where each pair of vertices represents an arm following a bernoulli distribution of mean $p$ or $q$. In our setting the minimal proportion $p^*$ would then be the proportion of pairs which belong to the same community and the gap as $\Delta = p - q$. The fundamental difference is that each arm can only be pulled once, making the problem significantly harder, however, the learner can exploit the SBM structure to their advantage. Assuming the case with exactly two equally sized communities with $T \leq |N|^2$, in [22] they show it is possible to attain a sub linear regret of the order $T\Delta \wedge \frac{(p+q)T}{\Delta}$. The significant worsening of their rate, in comparison to our own, is due to the fact one cannot sample an arm more than once, which significantly changes the flavour of their algorithms.

## 2 Cumulative regret

We first present an algorithm and prove an upper bound on its cumulative regret, and then we present a problem-dependent lower bound that shows we match the regret bound up to poly-log terms in $\Delta$. Lastly, we provide a theorem to the effect that adaptation to the proportion of optimal arms $p^\star$ is not possible in this setting.

### 2.1 Upper bound

We present `Sampling-UCB` for cumulative regret minimization. This algorithm is an Upper Confidence Bound (UCB) type algorithm [37]. We first sample a set $\mathcal{L}$ of arms large enough such that with high probability (of order $1 - 1/T$) there is a proportion of order $p^\star$ optimal arms. Then we build an upper confidence bound on the empirical mean of each sampled arm, see (2), where $\widehat{\mu}_a^t$ is the empirical mean of arm $a$ at time $t$ and $N_a^t$ the number of times arm $a$ was pulled until time $t$. At time $t$ we pull the arm $a \in \mathcal{L}$ with the highest upper confidence bound $U_a^t$. The complete procedure is detailed in Algorithm 1. Notably, we do not tune the upper confidence bounds such that they are exceeded with probability less than $1/T$, as for finite-armed bandits. In that setting, a common choice is to have bonuses of the form $\widehat{\mu}_a^t + \sqrt{2\log(T)/N_a^t}$, see [37]. Instead we use an exploration function that does not depend on $T$, such that the upper confidence bounds are exceeded with probability smaller than a fixed constant, see (2). Thus we only pay a constant regret of order $\log(1/\Delta)$ on the set of sampled arms $\mathcal{L}$. This is made possible by leveraging the fact that we know that there is a proportion of order $p^\star$ optimal arms.

**Input:** $\gamma \in (0, 1)$, $L \geq 1$
**Initialize:** Pick $\mathcal{L}$, with $|\mathcal{L}| = L$, arms from the reservoir $\mathcal{A}$. Sample each arm once.
**for** $t = L + 1$ *to* $T$ **do**
> Compute for each arm $a \in \mathcal{L}$ the quantity
>
> $$U_a^t = \widehat{\mu}_a^t + \sqrt{\frac{\gamma^2(1-\gamma)^{-1}/4 + \log(\pi^2/6) + 2\log(N_a^t)}{2N_a^t}}, \qquad (2)$$
>
> Play $a_t = \arg\max_{a \in \mathcal{L}} U_a^t$.
**end**

**Algorithm 1:** Sampling UCB

We prove the following regret bound for `Sampling-UCB` in Appendix A.

**Theorem 1.** *For $T \geq 2$, $\gamma \in (0, 1)$ and $L = \lceil 4\log(T)/(p^\star\gamma^2) \rceil$, the expected cumulative regret of* `Sampling-UCB` *is upper bounded as follows:*

$$\mathbb{E}R(T) \leq O\left(\frac{\log(T)\log(1/\Delta)}{p^\star\Delta}\right),$$

*see the end of the proof for a precise bound, i.e.* (3).

Note that this bound matches the lower bound of Theorem 2 of Section 2.2, for $T$ large enough and up to a $\log(1/\Delta)$ multiplicative factor. Also, $L$ can be calibrated with a lower bound on $p^\star$ instead of $p^\star$, but this lower bound will appear in the rate instead of $p^\star$.

**Remark 1.** Algorithm `Sampling-UCB` samples $L$ arms uniformly at random from the reservoir. What we mean by this is that each arm is pulled at random from $\mathcal{A}$ *independently from the other pulled arms*. In other words, by doing this, we potentially artificially create several independent copies of the same arm – which might seem counter-intuitive, but is formally not a problem.
What this anyway implies is that the case $|\mathcal{A}| \leq L$ is not a problem – with this idea of independent copies, we can pull more arms from the reservoir than the number $|\mathcal{A}|$ of arms.

**Remark 2.** Our algorithm is reminiscent of that of [26], which, as our own, uses a UCB which does not depend on the time horizon, but only on the number of times an arm has been pulled. However, they do so for different reasons, namely to adapt to the infinite time horizon of the fixed confidence setting.

## 2.2 Lower bound

We can prove an equivalent of the [36] lower bound for finite-armed bandits for our setting. The following theorem is proved in Appendix A.

**Theorem 2.** *Consider* $\Delta \in (0, 1/4)$ *and* $p^\star \in (0, 1/4]$. *For any bandit algorithm, there exists a bandit problem in* $\mathfrak{B}_{\Delta, p^\star}$ *such that*

$$\mathbb{E}R(T) \geq \min\left(\frac{1}{60} \frac{\max\{\log(\Delta^2 T/16), 0\}}{p^\star \Delta}, \sqrt{T}\right)$$

Note that if we consider the gap $\Delta$ and the proportion of optimal arms $p^\star$ as fixed and $T$ large in comparison, i.e. $\Delta \gg \sqrt{1/T}$, then our lower bound is of order $\log(T)/(p^\star \Delta)$. This is the problem-dependent regime that we consider in this paper. On the contrary, if $\Delta \approx \sqrt{1/T}$ then our lower bound is of order $\sqrt{T}$. This is rather the problem-independent regime studied by [16]. We can make a parallel between the lower bound in our setting and the one for finite-armed bandits. Indeed, if we consider that the proxy for the number of arms is $|\mathcal{A}| \sim 1/p^\star$ which implies that there is $p^\star |\mathcal{A}| \sim 1$ optimal arm, then we recover the problem-dependent lower bound of order $|\mathcal{A}| \log(T)/\Delta$, if there are $|\mathcal{A}| - 1$ sub-optimal arms with gap $\Delta$.

## 2.3 Impossibility of adapting to $p^\star$

The following theorem shows that in the setting of minimizing the cumulative regret, it is impossible to adapt to the proportion of optimal arms $p^\star$. The theorem is proved in Appendix A.

**Theorem 3.** *Let* $p^\star \leq \frac{1}{4}$ *and* $c > 0$ *such that* $T \geq 4\left(\frac{c \log(T)}{p^\star \Delta^2}\right)^2$. *For any bandit algorithm* $\mathfrak{A}$ *such that for all bandit problems in* $\mathfrak{B}_{\Delta, p^\star}$, *we have,*

$$\mathbb{E}R(T) \leq \frac{c \log(T)}{p^\star \Delta}$$

*one has that* $\forall q^\star \leq \frac{4p^\star}{c}$ *there exists a problem in* $\mathfrak{B}_{\Delta, q^\star}$ *such that*

$$\mathbb{E}R(T) \geq \frac{\sqrt{T}\Delta}{4} .$$

**Remark 3.** The `Sampling-UCB` algorithm takes a user defined parameter $\gamma$ (which can be taken as a universal constant) and $L$, which should be calibrated depending on (a lower bound on) $p^\star$. While this is necessary, it is important to not that none of the parameters requires knowledge of $\Delta$.

# 3 Best-arm identification

We present our `Elimination` algorithm for best-arm identification, together with an upper bound on the probability of outputting a sub-optimal arm; next we prove a lower bound, which is matched by our upper bound up to a $1/\log(T)$ factor in the exponential.

## 3.1 Upper bound

As its name suggests, the `Elimination` algorithm (summarized in Algorithm 2) works by successive elimination of arms − through the update at round $i$ of a set $\mathcal{A}_i$ − although with a twist. We begin by sampling approximately $T$ arms at the first round. Namely, we first select a set $\mathcal{A}_1$ of $|\mathcal{A}_1| = \lfloor \bar{c}T/\log T \rfloor$ arms taken at random from the reservoir, for some constant $\bar{c} > 0$. Then at each round we use a $T/\log T$ fraction of our budget to sample the arms in our set. And so at round $i$ we sample each arm in the set $\mathcal{A}_i$ a number of $t_i = \lfloor \bar{c}T/(|\mathcal{A}_i| \log T) \rfloor$. We then eliminate half of the arms based on the arms' empirical means − namely, we just keep the $\lfloor |\mathcal{A}_i|/2 \rfloor \vee 1$ arms in $\mathcal{A}_i$ that have highest empirical means − and introduce an additional number of arms sampled from the reservoir distribution − namely $\lfloor |\mathcal{A}_i|/4 \rfloor$ − such that the final size of our arm set is reduced by $\frac{3}{4}$. At the end of the budget, we have one arm remaining − due to the choices of $\bar{c}$ − which is the arm that we return. Note that Remark 1 applies here too so that it is not a problem if $|\mathcal{A}|$ is smaller than the number of arms required by the algorithm. Theorem 4 is proved in Appendix B.

**Input:** $\bar{c}$
set $i \leftarrow 1$
**while** $i < \log T / \bar{c}$ **do**
> Sample each arm in $\mathcal{A}_i$ a number $t_i = \lfloor \bar{c} T / (|\mathcal{A}_i| \log T) \rfloor$ of times and compute their empirical means $(\hat{\mu}_i(a))_{a \in \mathcal{A}_i}$
> Put in $\mathcal{A}_{i+1}$ the $1 \vee \lfloor |\mathcal{A}_i| / 2 \rfloor$ arms that have highest empirical means $(\hat{\mu}_i(a))_{a \in \mathcal{A}_i}$, and add on top of that $\lfloor |\mathcal{A}_i| / 4 \rfloor$ new arms taken at random from the reservoir
> $i \leftarrow i + 1$

**end**
Return any $\hat{a}_T$ in $\mathcal{A}_i$

**Algorithm 2:** `Elimination`

**Theorem 4.** *Set $\bar{c} = \log(4/3)$.* `Elimination` *satisfies*

$$\mathbb{P}(\hat{a}_T \in \mathcal{A}^\star) \geq 1 - 2\log(T)\exp\left(-c\frac{\Delta^2 p^\star T}{\log T}\right),$$

*where $c = \bar{c}/19200$*

**Remark 4.** `Elimination` works by discarding many sub-optimal arms and few optimal arms in each round, so that at the end, when just one arm remains, it is optimal with high probability. A key element is that `Elimination` adds *fresh arms* from the reservoir at each round. This is to ensure that our algorithm is adaptive to $p^\star, \Delta$, as ensured by Theorem 4. Whenever the arms in $\mathcal{A}_i$ are pulled less than about $\Delta^{-2}$ times, there is no guarantee on what happens when half of the arms are eliminated. Therefore, we have to make sure that when the algorithm arrives at a round $i$ such that $t_i \gtrsim \Delta^{-2}$, the proportion of optimal arms is of larger order than $p^\star$ with high enough probability. This is ensured by adding the fresh arms added from the reservoir. Note that for some arm distributions, we do not need to add fresh arms and the algorithm would function also by just halving at each step the number of arms. Indeed, in the case where all arms follow a Bernoulli distribution, in terms of preserving the proportion of optimal arms, one can prove that halving the set of arms according to the empirical means is no worse than random halving of the set. Thus, in this case, with high probability we increase the proportion of optimal arms at each step, without diminishing it. This is however specific to the case of Bernoulli distributions and some other parametric families, and it is an open question whether this would be true in general.

**Remark 5.** The successive halving strategy our algorithm for best-arm identification is based on was first introduced by [29], however, without the trick of adding fresh arms, as they didn't need to be adaptive to $p^\star$.

### 3.2 Lower bound

The following Theorem provides a lower bound on the probability of error for best arm identification in our setting. The proof of Theorem 5 can be found in Appendix B.

**Theorem 5.** *Consider $\Delta \in (0, 1/4)$ and $p^\star \in [0, 1/4]$. For any bandit algorithm, there exists a bandit problem in $\mathfrak{B}_{\Delta, p^\star}$ such that*

$$\mathrm{e}(T) \geq \frac{1}{4}\exp\left(-Tp^\star\frac{\Delta^2}{32}\right).$$

In proving the above theorem we essentially show that an agent cannot accurately distinguish between two cases: $\mu^* = \frac{1}{2}$ and $\mu^* = \frac{1}{2} + \Delta$. That is, we consider two reservoirs $\mathbf{R}_0$ and $\mathbf{R}_1$ where $\mu_0^* = \frac{1}{2}$ and $\mu_1^* = \frac{1}{2} + \Delta$. Using a coupling argument we bound the KL divergence between the distribution of samples collected on $\mathbf{R}_0$ and $\mathbf{R}_1$. The results then follows by application of Bretagnolle-Huber's inequality.

## 4 Experiments

We conduct a preliminary set of experiments to test the performance of our algorithms. Specifically, for cumulative regret we compare our `Sampling-UCB` to the QRM1 algorithm by [16] and the

SR algorithm by Zhu and Nowak [44]. For simple regret we compare our `Elimination` to the BUCB algorithm by [32]. In both cases our performance appears comparable to the literature. See Appendix D for details.

## 5    Conclusion and open questions

Classifying optimal learning rates on the continuous armed bandit problems with a proportion of optimal arms and general reservoir distribution has been a question of interest in the literature for some time, see [27]. Recent papers – [4] and [44], while focused on a slightly different setting, have considerably weaker results when applied to our setting. Therefore, we believe our results mark a significant improvement in the state of the art. An extension of our results would be to remove the $\log(1/\Delta)$ discrepancy between UB and LB for cumulative regret. However, this appears non-trivial and in particular we struggle to see how a UCB based strategy would achieve this tighter bound in the case of the cumulative regret. Another possibility for further work is an expansion of our setting. Consider the arm reservoir $\mathcal{A}$ partitioned into $K$ possible distributions, each with associated probability $p_k$. Let $k^* = \arg\max_{[K]} \mu_k$ and take gaps $(\Delta_k)_{[K]} = (\mu_{k^*} - \mu_k)_{[K]}$. One could then consider more detailed bounds, dependent on the sequence $((p_k, \Delta_k))_{[K]}$ as opposed to just $p^\star$ and the smallest gap. The main difficulty here would be to deal with the case where some $p_k$ are much smaller than the proportion $p^\star$ corresponding to the optimal arm.

**Acknowledgements**    The work of J. Cheshire is supported by the Deutsche Forschungsgemein­schaft (DFG) GRK 2297 MathCoRe. The work of P. Ménard is supported by the SFI Sachsen-Anhalt for the project RE-BCI. The work of A. Carpentier is partially supported by the Deutsche Forschungs­gemeinschaft (DFG) Emmy Noether grant MuSyAD (CA 1488/1-1), by the DFG - 314838170, GRK 2297 MathCoRe, by the FG DFG, by the DFG CRC 1294 'Data Assimilation', Project A03, by the Forschungsgruppe FOR 5381 „Mathematische Statistik im Informationszeitalter – Statisti­sche Effizienz und rechentechnische Durchführbarkeit", Project 02, by the Agence Nationale de la Recherche (ANR) and the DFG on the French-German PRCI ANR ASCAI CA 1488/4-1 "Aktive und Batch-Segmentierung, Clustering und Seriation: Grundlagen der KI" and by the UFA-DFH through the French-German Doktorandenkolleg CDFA 01-18 and by the SFI Sachsen-Anhalt for the project RE-BCI.

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
