# A Cumulative regret proofs

## A.1 Upper Bound

*Proof of Theorem 1.* We denote by $\mathcal{L}$ the set of arms sampled from the reservoir such that $|\mathcal{L}| = L$. We also denote by $\mathcal{L}^\star = \{a \in \mathcal{L} : a \in \mathcal{A}^\star\}$ the set of optimal arms in $\mathcal{L}$ and by $L^\star = |\mathcal{L}^\star|$ its cardinality. Note that these quantities are all random.

Because of the choice of $L = \lceil 4\log(T)/(p^\star \gamma^2) \rceil$, we know that with high probability there is at least a proportion of $\gamma p^\star$ optimal arms in $\mathcal{L}$. Precisely, if we denote this favorable event by $\mathcal{E} = \{L^\star/L \geq (1-\gamma)p^\star\}$ then by Chernoff's inequality (see Lemma 3), we have

$$\mathbb{P}(\mathcal{E}^c) = \mathbb{P}\left(L^\star/L < (1-\gamma)p^\star\right) \leq e^{-\frac{\gamma^2}{4}Lp^\star} \leq \frac{1}{T}.$$

We can decompose the regret given this event and its complement:

$$\mathbb{E}[R(T)] = \mathbb{E}\left[\sum_{a \in \mathcal{L}}(\mu^\star - \mu_a)\mathbb{E}[N_a^T|\mathcal{L}]\mathbb{1}_{\mathcal{E}}\right] + T\mathbb{P}(\mathcal{E}^c)$$

$$\leq \mathbb{E}\left[\sum_{a \in \mathcal{L}/\mathcal{L}^\star}\Delta_a\mathbb{E}[N_a^T|\mathcal{L}]\mathbb{1}_{\mathcal{E}}\right] + 1.$$

We now follow the classical proof of UCB-type strategies to upper-bound the number of times a sub-optimal is pulled. From now on, we fix a set of sampled arms $\mathcal{L}$. Fix an $a \in \mathcal{L} \setminus \mathcal{L}^\star$. We have

$$\mathbb{E}[N_a^T|\mathcal{L}] \leq 1 + \sum_{t=L+1}^{T} \mathbb{P}(\forall b \in \mathcal{L}^\star, U_{t-1}^b \leq \mu^\star|\mathcal{L}) + \mathbb{P}(a_t = a, U_{t-1}^a \geq \mu^\star|\mathcal{L}).$$

For the first term in the summation we use the fact that there are many optimal arms. Precisely, using Hoeffding's inequality, we have

$$\mathbb{P}(\forall b \in \mathcal{L}^\star, U_{t-1}^b \leq \mu^\star|\mathcal{L}) \leq \mathbb{P}\left(\forall b \in \mathcal{L}^\star, \exists n \in [T] : \widehat{\mu}_{b,n} \right.$$
$$\left. + \sqrt{\frac{\gamma^2(1-\gamma)^{-1}/4 + \log(\pi^2/6) + 2\log(n)}{2n}} \leq \mu^\star \middle| \mathcal{L}\right)$$
$$\leq \prod_{b \in \mathcal{L}^\star}\left(\sum_{n=1}^{T}\frac{1}{n^2}e^{-\gamma^2(1-\gamma)^{-1}/4 - \log(\pi^2/6)}\right)$$
$$= e^{-\frac{\gamma^2}{4}(1-\gamma)^{-1}L^\star}.$$

For the second term we proceed as usual. Let

$$n_0 = \inf\left\{n \in \mathbb{N} : \sqrt{\frac{\gamma^2(1-\gamma)^{-1}/4 + \log(\pi^2/6) + 2\log(n)}{2n}} \leq \Delta/2\right\}$$

be such that pulling any arm $a \in \mathcal{A}_{sub}$ more than $n_0$ times is a small probability event. Note that thanks to Lemma 4

$$n_0 \leq 4\frac{(1-\gamma)^{-1} + \log\left(24(1-\gamma)^{-1}/\Delta^2\right)}{\Delta^2} + 1.$$

Then, using again Hoeffding's inequality for an arm $a \in \mathcal{L} \setminus \mathcal{L}^\star$, we obtain

$$\sum_{t=L+1}^{T}\mathbb{P}(a_t = a, U_{t-1}^a \geq \mu^\star|\mathcal{L}) \leq \sum_{n=n_a+1}^{T}\mathbb{P}(\widehat{\mu}_{a,n} - \mu \geq \Delta/2) + n_0$$

$$\leq \sum_{n \geq 1}e^{-n\Delta^2/2} + n_0 \leq n_0 + \frac{2}{\Delta^2}.$$

Collecting the previous inequalities we can conclude for $T \geq 2$

$$\mathbb{E}[R(T)] \leq \mathbb{E}\left[\sum_{a \in \mathcal{L}/\mathcal{L}^\star} T e^{-\gamma^2(1-\gamma)^{-1}L^\star/4} \mathbb{1}_{\mathcal{E}} + 1 + \Delta n_0 + \frac{2}{\Delta}\right] + 1$$

$$\leq \mathbb{E}\left[\sum_{a \in \mathcal{L}/\mathcal{L}^\star} T e^{-\gamma^2 L/4} \mathbb{1}_{\mathcal{E}} + 1 + \Delta n_0 + \frac{2}{\Delta}\right] + 1$$

$$\leq L\left(2 + \Delta n_0 + \frac{2}{\Delta}\right) + 1$$

$$\leq \frac{8\log(T)}{p^\star \Delta \gamma^2}\left(10(1-\gamma)^{-1} + 4\log\left(24(1-\gamma)^{-1}/\Delta^4\right)\right) + 1. \tag{3}$$

$\square$

## A.2 Lower Bound

We denote by $\mathcal{B}\mathrm{er}(p)$ the Bernoulli distribution of parameter $p$. The Kullback-Leibler (KL) divergence between probability distributions $P$ and $Q$ is denoted by $\mathrm{KL}(P, Q)$. In particular, the KL divergence between two Bernoulli distributions $\mathcal{B}\mathrm{er}(p)$ and $\mathcal{B}\mathrm{er}(q)$ is

$$\mathrm{kl}(p, q) = \mathrm{KL}\left(\mathcal{B}\mathrm{er}(p), \mathcal{B}\mathrm{er}(q)\right) = p\log\left(\frac{p}{q}\right) + (1-p)\log\left(\frac{1-p}{1-q}\right).$$

*Proof of Theorem 2.* We fix a partition of the reservoir $\mathcal{A} = \mathcal{A}_1 \cup \mathcal{A}_2 \cup \mathcal{A}_3$ and set $p^\star$ the probability to sample an arm in $\mathcal{A}_1$, $\mathcal{A}_2$ and $1 - 2p^\star$ the probability to sample an arm in $\mathcal{A}_3$. We define two bandits problems associated with this reservoir. The bandit problem $\nu$ where the arms in $\mathcal{A}_1$ have probability distribution $\mathcal{B}\mathrm{er}(1/2)$, the arm in $\mathcal{A}_2$ and $\mathcal{A}_3$ have probability distribution $\mathcal{B}\mathrm{er}(1/2 - \Delta)$. The second bandit problem $\nu'$ is such that the arms in $\mathcal{A}_1$ have probability distribution $\mathcal{B}\mathrm{er}(1/2)$, the arms in $\mathcal{A}_2$ have probability distribution $\mathcal{B}\mathrm{er}(1/2 + \Delta)$ and the arms in $\mathcal{A}_3$ have probability distribution $\mathcal{B}\mathrm{er}(1/2 - \Delta)$. We denote by $\mathbb{E}_\nu$ respectively $\mathbb{E}_{\nu'}$ the expectation under the bandit problem $\nu$ respectively $\nu'$.

Let $N^T_{\mathcal{A}_i} = \sum_{t=1}^T \mathbb{1}_{\{a_t \in \mathcal{A}_i\}}$ be the number of times an arm in $\mathcal{A}_i$ is pulled. Note that since the arms in $\mathcal{A}_2$ and $\mathcal{A}_3$ are indistinguishable for the agent in the problem $\nu$, it holds

$$\mathbb{E}_\nu[N^T_{\mathcal{A}_2}] = \frac{p^\star}{1 - p^\star}\mathbb{E}_\nu[N^T_{\mathcal{A}_2} + N^T_{\mathcal{A}_3}].$$

Let $I^t$ be the information available by the agent at time $t$, i.e. the collection of collected rewards and arms pulled. We denote by $\mathbb{P}^{I^t}_\nu$ respectively $\mathbb{P}^{I^t}_{\nu'}$ the distribution of this random variable under the bandit problem $\nu$ respectively $\nu'$. Thanks to the chain rule and the above remark we can upper bound the Kullback-Leibler divergence between these two probability distributions

$$\mathrm{KL}(\mathbb{P}^{I^T}_\nu, \mathbb{P}^{I^T}_{\nu'}) = \mathrm{kl}(1/2 - \Delta, 1/2 + \Delta)\mathbb{E}_\nu[N^T_{\mathcal{A}_2}]$$

$$= \mathrm{kl}(1/2 - \Delta, 1/2 + \Delta)\frac{p^\star}{1 - p^\star}\mathbb{E}_\nu[N^T_{\mathcal{A}_2} + N^T_{\mathcal{A}_3}]$$

$$\leq 22 p^\star \Delta^2 \mathbb{E}_\nu[N^T_{\mathcal{A}_2} + N^T_{\mathcal{A}_3}] = 22 p^\star \Delta \mathbb{E}_\nu[R(T)], \tag{4}$$

where in the last inequality we used that $p^\star \leq 1/4$ and

$$\mathrm{kl}(1/2 - \Delta, 1/2 + \Delta) = 2\Delta\log\left(1 + \frac{2\Delta}{1/2 - \Delta}\right) \leq \frac{4\Delta^2}{1/2 - \Delta} \leq 16\Delta^2.$$

We assume that

$$\mathbb{E}_\nu[R(T)] = \Delta\left(T - \mathbb{E}_\nu[N^T_{\mathcal{A}_1}]\right) \leq \sqrt{T}, \qquad \mathbb{E}_{\nu'}[R(T)] = \Delta\mathbb{E}_{\nu'}[N^T_{\mathcal{A}_1}] + 2\Delta\mathbb{E}_{\nu'}[N^T_{\mathcal{A}_3}] \leq \sqrt{T},$$

otherwise the result is trivially true. In particular this implies that

$$1 - \sqrt{\frac{1}{\Delta^2 T}} \leq \frac{\mathbb{E}_\nu[N^T_{\mathcal{A}_1}]}{T} \qquad \frac{\mathbb{E}_{\nu'}[N^T_{\mathcal{A}_1}]}{T} \leq \sqrt{\frac{1}{\Delta^2 T}}. \tag{5}$$

Using the contraction of the entropy (see Garivier et al. [21]), the inequality $\mathrm{kl}(x, y) \geq x \log(1/y) - \log(2)$ then (5), we obtain

$$
\begin{aligned}
\mathrm{KL}(\mathbb{P}_\nu^{I^T}, \mathbb{P}_{\nu'}^{I^T}) &\geq \mathrm{kl}\big(\mathbb{E}_\nu[N_{\mathcal{A}_1}^T]/T, \mathbb{E}_{\nu'}[N_{\mathcal{A}_1}^T]/T\big) \\
&\geq \frac{\mathbb{E}_\nu[N_{\mathcal{A}_1}^T]}{T} \log\left(\frac{T}{\mathbb{E}_{\nu'}[N_{\mathcal{A}_1}^T]}\right) - \log(2) \\
&\geq \frac{1}{2}\left(1 - \sqrt{\frac{1}{\Delta^2 T}}\right)\log(\Delta^2 T) - \log(2) .
\end{aligned}
$$

The previous inequality with the fact that the Kullback-Leibler divergence is positive yields

$$
\mathrm{KL}(\mathbb{P}_\nu^{I^T}, \mathbb{P}_{\nu'}^{I^T}) \geq \frac{2}{3} \log(\Delta^2 T/16)^+ . \tag{6}
$$

Indeed if $\Delta^2 T/16 \leq 1$ then (6) is trivially true. In the other case we have

$$
\begin{aligned}
\frac{1}{2}\left(1 - \sqrt{\frac{1}{\Delta^2 T}}\right)\log(\Delta^2 T) - \log(2) &\geq \frac{3}{8}\log(\Delta^2 T) - \frac{1}{4}\log(16) \\
&\geq \frac{3}{8}\log(\Delta^2 T/16)
\end{aligned}
$$

Combining (4) and (6) allows us to conclude

$$
\mathbb{E}_\nu\big[R(T)\big] \geq \frac{1}{60} \frac{\log(\Delta^2 T/16)^+}{p^\star \Delta}.
$$

$\square$

### A.3 Impossibility of adaptation to $p^\star$

*Proof of Theorem 3.* Consider $\Delta \in (0, 1/4)$ and the following two definitions of two reservoir distributions:

- The reservoir distribution $\mathbf{R}_0$ characterised by $p_1 = p^\star$ and $p_2 = 1 - p^\star$ and $\nu_1 = \mathcal{B}(1/2)$ and $\nu_2 = \mathcal{B}(1/2 - \Delta)$.

- The reservoir distribution $\mathbf{R}_1$ characterised by $p_1 = q^\star$, $p_2 = p^\star$ and $p_3 = 1 - q^\star - p^\star$ and $\nu_1 = \mathcal{B}(1/2 + \Delta)$ and $\nu_2 = \mathcal{B}(1/2)$ and $\nu_3 = \mathcal{B}(1/2 - \Delta)$.

Note that the Bernoulli distribution is completely characterised by its mean and so we can use the mean to characterise the distribution. Let $\tilde{\mu} = (\tilde{\mu}_j)_{j \leq T}$ be $T$ i.i.d. means corresponding to $T$ i.i.d. distributions sampled according to the reservoir distribution $\mathbf{R}_1$. Note that $\tilde{\mu}_j \in \{1/2 - \Delta, 1/2, 1/2 + \Delta\}$. Write also $\tilde{\mu}' = (\tilde{\mu}'_j)_{j \leq T}$ for the vector of means such that $\tilde{\mu}'_j = \tilde{\mu}_j$ if $\tilde{\mu}'_j \in \{1/2 - \Delta, 1/2\}$, and $\tilde{\mu}'_j = 1/2 - \Delta$ otherwise. Note that then, we have that $(\tilde{\mu}'_j)_{j \leq T}$ are $T$ i.i.d. means corresponding to $T$ i.i.d. distributions sampled according to the reservoir distribution $\mathbf{R}_0$, by definition of $\mathbf{R}_0$. Write $\mathbb{E}_{\mathbf{R}_1}$ for the expectation according to the distribution of $\tilde{\mu}$, i.e. according to $\mathbf{R}_1^{\otimes T}$, and $\mathbb{E}_{\mathbf{R}_0}$ for the expectation according to the distribution of $\tilde{\mu}'$, i.e. according to $\mathbf{R}_0^{\otimes T}$.

Consider an algorithm $\mathfrak{A}$ and a bandit problem involving Bernoulli distributions characterised by a vector of means $m = (m_j)_{j \leq T}$. Write $\mathbb{P}_m^{\mathfrak{A}}$ for the distribution of the samples obtained by the algorithm run on this problem, and $\mathbb{E}_m^{\mathfrak{A}}$ the associated expectation. Consider now another Bernoulli bandit problem characterised by the means $m' = (m'_j)_{j \leq T}$. We have because of the chain rule

$$
\mathrm{KL}(\mathbb{P}_{m'}^{\mathfrak{A}}, \mathbb{P}_m^{\mathfrak{A}}) = \sum_{j \leq T} \mathbb{E}_{m'}^{\mathfrak{A}}[T_j] \, \mathrm{kl}(m'_j, m_j),
$$

where $\mathbb{E}_{m'}^{\mathfrak{A}}$ is the expectation according to problem $m'$ on which algorithm $\mathfrak{A}$ is used, and where $T_j$ is the number of times arm $j$ is sampled at time $T$.

From our assumption on $\mathfrak{A}$ we have that $\mathbb{E}_{\mathbf{R}_0}[R(T)] \leq \frac{\log(T)}{p^\star \Delta}$. Now, we can obtain

$$
\begin{aligned}
\mathrm{KL}(\mathbb{E}_{\mathbf{R}_0}\mathbb{P}^{\mathcal{A}}_{\tilde{\mu}'}, \mathbb{E}_{\mathbf{R}_1}\mathbb{P}^{\mathcal{A}}_{\tilde{\mu}}) = {} & \mathrm{KL}(\mathbb{E}_{\mathbf{R}_1}\mathbb{P}^{\mathfrak{A}}_{\tilde{\mu}'}, \mathbb{E}_{\mathbf{R}_1}\mathbb{P}^{\mathfrak{A}}_{\tilde{\mu}}) \\
\leq {} & \mathbb{E}_{\mathbf{R}_1}\left[ \mathrm{KL}(\mathbb{P}^{\mathfrak{A}}_{\tilde{\mu}'}, \mathbb{P}^{\mathfrak{A}}_{\tilde{\mu}}) \right] = \mathbb{E}_{\mathbf{R}_1}\left[ \sum_{j\leq T} \mathbb{E}^{\mathfrak{A}}_{\tilde{\mu}'}[T_j]\,\mathrm{kl}(\tilde{\mu}'_j, \tilde{\mu}_j) \right] \\
\leq {} & \mathbb{E}_{\mathbf{R}_1}\left[ \sum_{j\leq T} \mathbb{E}^{\mathfrak{A}}_{\tilde{\mu}'}[T_j]\frac{\Delta^2}{16}\mathbf{1}\{\tilde{\mu}_j = 1/2 + \Delta\} \right] \\
= {} & \mathbb{E}_{\mathbf{R}_0}\left[ \sum_{j\leq T} \mathbb{E}_{\tilde{\mu}', \mathfrak{A}}[T_j]\frac{\Delta^2}{16}\mathbf{1}\{\tilde{\mu}'_j = 1/2 - \Delta\}\frac{q^\star}{1 - p^\star} \right] \\
= {} & \frac{q^\star \Delta}{8}\mathbb{E}_{\mathbf{R}_0}[R(T)] \leq \frac{cq^\star}{8p^\star}\log(T) \leq \frac{1}{2}\log(T), \qquad (7)
\end{aligned}
$$

where the last equality follows since by definition of $\mathbf{R}_0, \mathbf{R}_1$, conditionally on $\tilde{\mu}'_j = 1/2 - \Delta$, the probability that $\tilde{\mu}_j = 1/2 + \Delta$ is $\frac{q^\star}{1-p^\star} \leq 2q^\star$, and otherwise it is 0. And where the final inequality comes from our assumption $p^\star > \frac{cq^\star}{4}$.

Consider the event,

$$
E := \left\{ \sum_{j\leq T} T_j\mathbf{1}\{\tilde{\mu}'_j = 1/2\} > T/2 \right\}.
$$

Note that on $\mathbf{R}_0$, we have $\mu^* = \frac{1}{2}$. Thus, on $\mathbf{R}_0$ the event $E^C$ will signify a regret greater than $\frac{T\Delta}{2}$, similarly on $\mathbf{R}_1$ the event $E$ signifies a regret greater than $\frac{T\Delta}{2}$. Thus,

$$
E^C \subset \left\{ R_{\mathbf{R}_0}(T) \geq \frac{T\Delta}{2} \right\}, \qquad E \subset \left\{ R_{\mathbf{R}_1}(T) \geq \frac{T\Delta}{2} \right\}. \qquad (8)
$$

Where $R_{\mathbf{R}_0}(T)$ and $R_{\mathbf{R}_1}(T)$ denote the regret of the algorithm on $\mathbf{R}_0$ and $\mathbf{R}_1$ respectively. Now from our assumption upon $\mathfrak{A}$ we have that $\mathbb{E}_{\mathbf{R}_0}R(T) \leq \frac{c\log(T)}{p^\star \Delta}$, therefore Equation (8) leads to,

$$
\mathbb{E}_{\mathbf{R}_0}\mathbb{P}^{\mathfrak{A}}_{\tilde{\mu}'}\big(E^C\big) \leq \frac{c\log(T)}{p^\star \Delta} \times \frac{2}{T\Delta}. \qquad (9)
$$

and in addition we also have,

$$
\mathbb{E}_{\mathbf{R}_1}R(T) \geq \mathbb{E}_{\mathbf{R}_1}\mathbb{P}^{\mathfrak{A}}_{\tilde{\mu}}(E) \times \frac{T\Delta}{2}. \qquad (10)
$$

Now, using the Bretagnolle-Huber's inequality (see Theorem 14.2 by Lattimore and Szepesvári [37]) in combination with (7) we obtain

$$
\begin{aligned}
\mathbb{E}_{\mathbf{R}_0}\mathbb{P}^{\mathfrak{A}}_{\tilde{\mu}'}(E^C) + \mathbb{E}_{\mathbf{R}_1}\mathbb{P}^{\mathfrak{A}}_{\tilde{\mu}}(E) \geq {} & \frac{1}{2}\exp\bigg( - \mathrm{KL}(\mathbb{E}_{\mathbf{R}_1}\mathbb{P}^{\mathfrak{A}}_{\tilde{\mu}'}, \mathbb{E}_{\mathbf{R}_1}\mathbb{P}^{\mathfrak{A}}_{\tilde{\mu}}) \bigg) \\
\geq {} & \frac{1}{2\sqrt{T}}.
\end{aligned}
$$

This result in combination with Equation (9) gives the following,

$$
\mathbb{E}_{\mathbf{R}_1}\mathbb{P}^{\mathfrak{A}}_{\tilde{\mu}}(E) \geq \frac{1}{2\sqrt{T}} - \frac{2c\log(T)}{p^\star T\Delta^2} \geq \frac{1}{4\sqrt{T}} \qquad (11)
$$

where the final inequality comes from our assumption $T \geq 4\left(\frac{c\log(T)}{p^\star \Delta^2}\right)^2$. Finally our result follows from combination of Equation (9) and Equation (11).

$\square$

# B   Best-arm identification proofs

## B.1   Upper Bound

*Proof of Theorem 4.* **Proof-specific notations and preliminary considerations.** At round $i$, write $K_i = |\mathcal{A}_i|$ and write $p_i$ for the proportion of optimal arms in $\mathcal{A}_i$, namely

$$p_i = |\mathcal{A}_i \cap \mathcal{A}^*|/|\mathcal{A}_i|.$$

We also write $M_i$ for the number of optimal arms in $\mathcal{A}_i$ such that $\hat{\mu}_i(a) \geq \mu^* - \Delta/2$, namely

$$M_i = \left|\{a \in \mathcal{A}_i \cap \mathcal{A}^* : \hat{\mu}_i(a) \geq \mu^* - \Delta/2\}\right|,$$

and $N_i$ for the number of sub-optimal arms in $\mathcal{A}_i$ such that $\hat{\mu}_i(a) \geq \mu^* - \Delta/2$, namely

$$N_i = \left|\{a \in \mathcal{A}_i \cap \mathcal{A}_{sub} : \hat{\mu}_i(a) \geq \mu^* - \Delta/2\}\right|.$$

Note that by definition

$$K_{i+1} = \left(1 \vee \left\lfloor \frac{K_i}{2} \right\rfloor\right) + \left\lfloor \frac{K_i}{4} \right\rfloor.$$

Therefore the following bounds holds

$$\left(\left(\frac{3}{4}\right)^i K_1\right) \vee 1 \geq K_i \geq \left(\frac{1}{2}\right)^i K_1 - 4. \tag{12}$$

We write $I$ for the smallest index $i$ such that $K_i = 1$ and will not investigate what happens at rounds $i > I$. By the upper bound (12) on $K_i$ it holds $I \leq \log_{4/3}(K_1) \leq \log_{4/3}(T)$. Note that since $\log_{4/3}(T) = \bar{c} \log T$, the algorithm terminates with a set containing just one arm.

**Step 1: Introduction of high-probability events of interest.** We define the constant

$$c = \frac{\bar{c}}{10}.$$

We define $j^*$ as the largest $j$ smaller than or equal to $I$ such that

$$K_j \geq cT\Delta^2/(2\log T).$$

Note that such $j^*$ exists since $K_1 \geq \bar{c}T/(2\log T)$, and since $K_I = 1$. We prove below the following upper bound on $j^*$. Take any round $i$. Note that for any $k$, conditionally on $\mathcal{A}_i$, by Hoeffding's inequality, for any $a \in \mathcal{A}_i$

$$\mathbb{P}\left(\left|\hat{\mu}_i(a) - \mu_i(a)\right| \geq \Delta/2 \,\middle|\, \mathcal{A}_i\right) \leq 2\exp(-\Delta^2 t_i/2) = q_i, \tag{13}$$

where $\mu_i(a)$ is the true mean associated with arm $a$. We now state the following technical lemma proved below.

**Lemma 1.** *Assume that $p^\star \leq 1/2$, and consider $I \geq i \geq j^*$. Under the assumptions of the theorem, we have*

$$q_i^{-1/2} \geq 200 \geq e^2 - 1, \tag{14}$$
$$\Delta^2 t_i/4 \geq \log 2. \tag{15}$$

We define for $i \geq j^*$ and $\bar{p}_i := \left(\frac{p^\star}{6}(5/4)^{i-j^*} \wedge (1/2)\right)$, the event

$$\xi_i = \{p_i > \bar{p}_i\}.$$

Consider from now on $i \geq j^*$.

**Step 2: Lower bound on $M_i$ conditional to $\xi_i$.** We have by definition of $M_i$:

$$M_i = \sum_{a \in \mathcal{A}_i \cap \mathcal{A}^*} \mathbf{1}\{\hat{\mu}_i(a) \geq \mu^* - \Delta/2\},$$

where by Equation (13), and conditionally on $\mathcal{A}_i$, the $\mathbf{1}\{\hat{\mu}_i(a) \geq \mu^* - \Delta/2\}$ are independent and dominate stochastically $\mathcal{B}(1 - q_i)$, for any $a \in \mathcal{A}_i \cap \mathcal{A}^*$. And so conditionally on $\mathcal{A}_i$, we have that $M_i$ stochastically dominates $\mathcal{B}(K_i p_i, 1 - q_i)$. And so by Chernoff's inequality, for any $x \geq \sqrt{q_i}$:

$$\mathbb{P}(M_i - p_i K_i(1 - q_i) \leq -x p_i K_i | \mathcal{A}_i) \leq \left[ \frac{e^{x/q_i}}{(1 + x/q_i)^{1+x/q_i}} \right]^{K_i p_i q_i}$$
$$\leq \exp\left[ x K_i p_i - \log(1 + x/q_i)(K_i p_i q_i + x K_i p_i) \right]$$
$$\leq (1 + x/q_i)^{-x K_i p_i / 2}.$$

as for $i > j^*$ we have $\log(1 + x/q_i) > 2$, see Lemma 1.

So that for $x \geq \sqrt{q_i}$

$$\mathbb{P}(M_i \leq K_i p_i(1 - 2x) | \mathcal{A}_i) \leq \exp\left( -x \Delta^2 t_i K_i p_i / 16 \right),$$

since $\log(q_i^{-1}) = \Delta^2 t_i / 2 - \log 2 \geq \Delta^2 t_i / 4$ for $I \geq i \geq j^*$ - see Lemma 1.

And so since $p_i \geq \frac{p^\star}{6}$ on $\xi_i$

$$\mathbb{P}(M_i \leq p_i K_i(1 - 2x) | \xi_i) \leq \exp\left( -\bar{c}' x p^\star \Delta^2 T / \log T \right) := u. \tag{16}$$

where $\bar{c}' = \bar{c}/96$ and recalling $t_i = \lfloor \bar{c} T / (K_i \log(T)) \rfloor$.

**Step 3: Upper bound on $N_i$ conditional to $\xi_i$.** We have by definition of $N_i$:

$$N_i = \sum_{a \in \mathcal{A}_i \cap \mathcal{A}_{sub}} \mathbf{1}\{\hat{\mu}_i(a) \geq \mu^* - \Delta/2\},$$

where by Equation (13), and conditionally on $\mathcal{A}_i$, the $\mathbf{1}\{\hat{\mu}_i(a) \geq \mu^* - \bar{\Delta}/2\}$ are independent and are stochastically dominated by $\mathcal{B}(q_i)$, for any $a \in \mathcal{A}_i \cap \mathcal{A}_{sub}$. And so conditionally on $\mathcal{A}_i$, we have that $N_i$ is stochastically dominated by $\mathcal{B}(K_i, q_i)$. And so by Chernoff's inequality for any $x \geq 2$:

$$\mathbb{P}(N_i - K_i q_i \geq x K_i | \xi_i) \leq \left[ \frac{e^{x/q_i}}{(1 + x/q_i)^{1+x/q_i}} \right]^{K_i q_i} \leq (1 + x/q_i)^{-x K_i / 2},$$

similar to Step 2.

So that for $x \geq \sqrt{q_i}$

$$\mathbb{P}(N_i \geq 2 K_i x | \mathcal{A}_i) \leq \exp\left( -x \Delta^2 t_i K_i / 16 \right),$$

as in Step 2.

And so similar to in Step 2:

$$\mathbb{P}(N_i \geq 2 x K_i | \xi_i) \leq \exp\left( -\bar{c}' x \Delta^2 T / \log T \right) \leq u. \tag{17}$$

**Step 4: Bound on the probability of $\xi_i$ and conclusion.** First we have – since we add $K_{j^*-1}/4 = K_{j^*}/3$ fresh arms to the set $\mathcal{A}_{j^*}$ - that

$$\left\{ \left| \sum_{a \in \mathcal{A}_{j^*}} \mathbf{1}\{a \in \mathcal{A}^*\} - \frac{1}{3} p^\star K_{j^*} \right| \leq \frac{1}{6} p^\star K_{j^*} \right\} \subset \xi_{j^*},$$

where it holds that $\mathbf{1}\{a \in \mathcal{A}^*\} \sim \mathcal{B}(p^*)$ for the fresh arms and $|\mathcal{A}_{j^*}| = K_{j^*}$. And so by Chernoff's inequality:

$$\mathbb{P}(\xi_{j^*}) \geq 1 - 2\exp(-p^\star K_{j^*}/10) \geq 1 - 2\exp\left( -c \frac{p^\star T \Delta^2}{20 \log T} \right) =: 1 - v, \tag{18}$$

by definition of $j^*$.

Now consider $i > j^*$, let,

$$\xi_i' = \left\{ p_{i+1} \geq \frac{5}{4} p_i \wedge \frac{1}{2} \right\}.$$

**Lemma 2.** *Assume that $2x \leq 1/100$. We have for $I \geq i > j^*$:*

$$\xi_i'' := \{ M_i > p_i K_i (1 - 2x) \} \cap \{ N_i < 2x K_i \} \subset \xi_i'.$$

Note also that

$$\mathbb{P}(\xi_i'' | \xi_i) \geq 1 - 2u,$$

by Equations (16) and (17), so that by Lemma 2

$$\mathbb{P}(\xi_i' | \xi_i) \geq 1 - 2u. \tag{19}$$

By induction it holds that for any $1 \leq m \leq I - j^*$

$$\xi_{j^*} \cap \bigcap_{j^* < i \leq j^* + m} \xi_i' \subset \bigcap_{j^* \leq i \leq j^* + m} \xi_i,$$

so that by Equations (18) and (19)

$$\mathbb{P}\left( \bigcap_{j^* \leq i \leq j^* + m} \xi_i \right) \geq (1 - v)(1 - 2u)^m \geq 1 - v - 2um.$$

In particular using the previous inequality for $m = I - j^*$ and since $I \leq \log T$ it holds

$$\mathbb{P}\left( \bigcap_{j^* \leq i \leq I} \xi_i \right) \geq 1 - v - 2u \log T.$$

Since $K_I = 1$, and since by definition of the $\xi_i$ we know that on $\xi_I$ we have that the only arm in $\mathcal{A}_I$ is optimal, this concludes the proof - taking $x = 1/200$, which is compatible with $x \geq \sqrt{q_i}$ as $q_i \leq 1/200^2$ by Lemma 1.

$\square$

We prove now successively, Lemma 1, Lemma 2 used in the proof of Theorem 4.

*Proof of Lemma 1.* Note first that for $I \geq i \geq j^*$ we have

$$K_{i+1} = \lfloor K_i/2 \rfloor \vee 1 + \lfloor K_i/4 \rfloor \leq \frac{3K_i}{4} \vee 1.$$

So that for any $0 \leq m < I - j^*$ we have by definition of $I$ as the first index such that $K_I = 1$

$$K_i \leq K_{j^*}(3/4)^m. \tag{20}$$

Also for any $i$ such that $K_i \geq 4$

$$K_{i+1} \geq K_i/2,$$

and for any $i$ such that $K_i < 4$, we have

$$K_{i+1} = 1,$$

so that for any $0 \leq m < I - j^*$ we have

$$K_i \geq K_{j^*}(i/2)^m.$$

**Inequality (14):** We therefore have for $I > i \geq j^*$ and by Equation (20)

$$q_i^{-1/2} = 2^{-1/2} \exp(\Delta^2 t_i/4) \geq 2^{-1/2} \exp\left( \bar{c} \frac{\Delta^2 T}{2 K_{j^*} \log(T)} \right),$$

$$\geq 2^{-1/2} \exp(10) \geq 200 \geq e^2 - 1$$

**Inequality** (15): We have,
$$q_i = \exp(-\Delta^2 t_i/2) \,,$$
thus by inequality (14) we have
$$\exp(\Delta^2 t_i/4) \geq \sqrt{2}(e^2 - 1),$$
so that
$$\Delta^2 t_i/4 \geq \log 2.$$

$\square$

*Proof of Lemma 2.* Let $i$ such that $I \geq i > j^*$. Note that on $\xi_i''$, we have $M_i > 0$ so that $p_i > 0$.

**First case:** $0 < p_i \leq 2/5$. Assume first that $p_i \leq 2/5$. On $\xi_i''$ we have that
$$M_i > p_i K_i (1 - 2x),$$
and
$$N_i < 2K_i x,$$
so that
$$M_i + N_i < p_i K_i + 2K_i x \leq (2/5)K_i + K_i/100 \leq K_i/2.$$
since $2x \leq 1/100$ for $i \geq j^*$ - see Lemma 1. And so all $M_i$ arms of $\{a \in \mathcal{A}_i \cap \mathcal{A}^* : \hat{\mu}_i(a) \geq \mu^* - \bar{\Delta}/2\}$ are going to be in $\mathcal{A}_{i+1}$. This implies – as in this case $K_i \geq 2$ otherwise we cannot have $0 < p_i \leq 2/5$ – that
$$p_{i+1} \geq \frac{M_i}{K_{i+1}} = \frac{M_i}{1 \vee \lfloor K_i/2 \rfloor + \lfloor K_i/4 \rfloor} \geq \frac{4}{3}(1 - 2x)p_i > \frac{5}{4}p_i,$$
as $2x \leq 1/100$.

**Second case:** $p_i > 2/5$. Assume now that $p_i > 2/5$. On $\xi_i''$ we have that
$$M_i > p_i K_i (1 - 2x) \geq \frac{198}{500}K_i,$$
and
$$N_i < 2K_i x \leq K_i/100,$$
since $2x \leq 1/100$ for $I \geq i > j^*$ – see Lemma 1. Since $198/500 + 1/100 = 203/500 < 1/2$ this implies that at least $\frac{199}{500}K_i$ from the arms in $\{a \in \mathcal{A}_i \cap \mathcal{A}^* : \hat{\mu}_i(a) \geq \mu^* - \bar{\Delta}/2\}$ are going to be in $\mathcal{A}_{i+1}$. So that
$$p_{i+1} \geq \frac{M_i}{K_{i+1}} = \frac{M_i}{1 \vee \lfloor K_i/2 \rfloor + \lfloor K_i/4 \rfloor} \geq \frac{4}{3} \times \frac{198}{500} = \frac{66}{125} > 1/2.$$

This concludes the proof. $\square$

## B.2 Lower Bound

*Proof of Theorem 5.* We consider a similar setting to that in the proof of Theorem 3 although with a slightly different construction of $\mathbf{R}_0, \mathbf{R}_1$.

Consider the following two reservoir distributions:

- The reservoir distribution $\mathbf{R}_0$ characterised by $p_1 = p^\star$ and $p_2 = 1 - p^\star$ and $\nu_1 = \mathcal{B}(1/2)$ and $\nu_2 = \mathcal{B}(1/2 - \Delta)$.

- The reservoir distribution $\mathbf{R}_1$ characterised by $p_1 = p^\star$ and $p_2 = p^\star$ and $p_3 = 1 - 2p^\star$ and $\nu_1 = \mathcal{B}(1/2 + \Delta)$ and $\nu_2 = \mathcal{B}(1/2)$ and $\nu_3 = \mathcal{B}(1/2 - \Delta)$.

We define $\tilde{\mu}, \tilde{\mu}'$, and associated expectations and probabilities as in the proof of Theorem 3. Consider also any algorithm $\mathfrak{A}$. We have by similar calculations as Equation (7) the following upper bound on the KL divergence

$$\mathrm{KL}(\mathbb{E}_{\mathbf{R}_0}\mathbb{P}_{\tilde{\mu}'}^{\mathfrak{A}}, \mathbb{E}_{\mathbf{R}_1}\mathbb{P}_{\tilde{\mu}}^{\mathfrak{A}}) = \mathrm{KL}(\mathbb{E}_{\mathbf{R}_1}\mathbb{P}_{\tilde{\mu}'}^{\mathfrak{A}}, \mathbb{E}_{\mathbf{R}_1}\mathbb{P}_{\tilde{\mu}}^{\mathfrak{A}})$$

$$\leq \mathbb{E}_{\mathbf{R}_1}\left[\mathrm{KL}(\mathbb{P}_{\tilde{\mu}'}^{\mathfrak{A}}, \mathbb{P}_{\tilde{\mu}}^{\mathfrak{A}})\right] = \mathbb{E}_{\mathbf{R}_1}\left[\sum_{j \leq T}\mathbb{E}_{\tilde{\mu}'}^{\mathfrak{A}}[T_j]\,\mathrm{kl}(\tilde{\mu}'_j, \tilde{\mu}_j)\right]$$

$$\leq \mathbb{E}_{\mathbf{R}_1}\left[\sum_{j \leq T}\mathbb{E}_{\tilde{\mu}'}^{\mathfrak{A}}[T_j]\frac{\Delta^2}{16}\mathbf{1}\{\tilde{\mu}_j = 1/2 + \Delta\}\right]$$

$$= \mathbb{E}_{\mathbf{R}_0}\left[\sum_{j \leq T}\mathbb{E}_{\tilde{\mu}'}^{\mathfrak{A}}[T_j]\frac{\Delta^2}{16}\mathbf{1}\{\tilde{\mu}'_j = 1/2 - \Delta\}\frac{p^\star}{1 - p^\star}\right], \qquad (21)$$

since by definition of $\mathbf{R}_0, \mathbf{R}_1$, conditionally on $\tilde{\mu}'_j = 1/2 - \Delta$, the probability that $\tilde{\mu}_j = 1/2 + \Delta$ is $\frac{p^\star}{1 - p^\star}$, and otherwise it is 0.

By Equation (21) and since $\sum_{j \leq T}\mathbb{E}_{\tilde{\mu}'}^{\mathfrak{A}}[T_j] = T$, we have

$$\mathrm{KL}(\mathbb{E}_{\mathbf{R}_0}\mathbb{P}_{\tilde{\mu}'}^{\mathfrak{A}}, \mathbb{E}_{\mathbf{R}_1}\mathbb{P}_{\tilde{\mu}}^{\mathfrak{A}}) \leq T\frac{\Delta^2}{16}\frac{p^\star}{1 - p^\star}.$$

Now by Bretagnolle-Huber's inequality (see Theorem 14.2 by Lattimore and Szepesvári [37]) and for any event $E$

$$\mathbb{E}_{\mathbf{R}_1}\mathbb{P}_{\tilde{\mu}}^{\mathfrak{A}}(E) + \mathbb{E}_{\mathbf{R}_0}\mathbb{P}_{\tilde{\mu}'}^{\mathfrak{A}}(E^C) \geq \frac{1}{2}\exp\left(-\mathrm{KL}(\mathbb{E}_{\mathbf{R}_0}\mathbb{P}_{\tilde{\mu}'}^{\mathfrak{A}}, \mathbb{E}_{\mathbf{R}_1}\mathbb{P}_{\tilde{\mu}}^{\mathfrak{A}})\right). \qquad (22)$$

Let us write $\hat{a}_T$ for the arm that the algorithm $\mathfrak{A}$ recommends. Set

$$E = \{\tilde{\mu}_{\hat{a}_T} = 1/2\}.$$

Note that on $E$, we make a mistake in prediction for $\tilde{\mu}$, and that on $E^C$, we make a mistake in prediction for $\tilde{\mu}'$. We have

$$\mathbb{E}_{\mathbf{R}_1}\mathbb{P}_{\tilde{\mu}}^{\mathfrak{A}}(E) + \mathbb{E}_{\mathbf{R}_1}\mathbb{P}_{\tilde{\mu}'}^{\mathfrak{A}}(E^C) \geq \frac{1}{2}\exp\left(-T\frac{\Delta^2}{16}\frac{p^\star}{1 - p^\star}\right).$$

This concludes the proof by definition of $E$. $\qquad\qquad\square$

## C   Technical lemmas

**Lemma 3.** *(Chernoff bound) Let $X_1, \ldots, \mathcal{X}_n \sim \mathcal{B}\mathrm{er}(p)$ be $n$ samples from a Bernoulli distribution and $S_n = \sum_{k=1}^n X_n$ their sum. Then for all $\gamma \in [0, 1]$ it holds*

$$\mathbb{P}\left(\frac{S_n}{n} \leq (1 - \gamma)p\right) \leq e^{-\frac{\gamma^2}{4}np},$$

$$\mathbb{P}\left(\frac{S_n}{n} \geq (1 + \gamma)p\right) \leq e^{-\frac{\gamma^2}{4}np}.$$

*Proof.* We prove the first inequality; the second one is similar. If $(1 - \gamma)p < 0$ or $\gamma = 0$ the inequality is trivially true. Else, because of Chernoff's inequality, we have

$$\mathbb{P}\left(\frac{S_n}{n} \leq (1 - \gamma)p\right) \leq e^{-n\,\mathrm{kl}\left((1-\gamma)p, p\right)}.$$

It remains to remark to conclude that

$$\mathrm{kl}\left((1 - \gamma)p, p\right) \geq \frac{\gamma^2}{2}p,$$

where we used the refined Pinsker inequality from Garivier et al. [21], for $0 \leq x < y \leq 1$,

$$\mathrm{kl}(y, x) \geq \frac{1}{2 \max_{x \leq q \leq y} q(1-q)} (x-y)^2 \geq \frac{1}{2y}(x-y)^2 \, .$$

For the second inequality we use

$$\mathrm{kl}\left((1+\gamma)p, p\right) \geq \frac{1}{2(1+\gamma)p}\gamma^2 p^2 \geq \frac{\gamma^2}{4}p \, .$$

$\square$

**Lemma 4.** *Let $A, B, C \geq 0$ be constants such that $A \geq C$, then for $n_0 = \inf\{n \geq 1 : A + B\log(n) \leq nC\}$ we have*

$$n \leq \frac{A + B\log\left((2(B^2 + AC)/C^2\right)}{C} + 1 \, .$$

*Proof.* First let $x_0 \geq 1$ be such that $A + B\log(x_0) = Cx_0$. It exists since $A + B\log(x)/x \to 0$ if $x \to \infty$ and since $A \geq C$. In particular, because of the definition of $n_0$ we have $x_0 \leq n_0 \leq x_0 + 1$. Then note that $A + B\sqrt{x_0} \leq Cx_0$. Thus $\sqrt{x_0}$ is smaller than the largest roots of the polynomial $Cy^2 - By - A$. Using $\sqrt{a+b} \leq \sqrt{a} + \sqrt{b}$ and $(a+b)^2 \leq 2(a^2 + b^2)$ we obtain

$$x_0 \leq \left(\frac{B + \sqrt{B^2 + 4AC}}{2C}\right)^2$$

$$\leq 2\frac{B^2 + AC}{C^2} \, .$$

Inserting the previous inequality in the definition of $x_0$ and using $n_0 \leq x_0 + 1$ allows us to conclude

$$n_0 \leq \frac{A + B\log\left(2(B^2 + AC)/C^2\right)}{C} + 1 \, .$$

$\square$

## D    Experiments

In this section we conduct preliminary experiments for the cumulative regret and best-arm identification setting.

**Cumulative regret**    For the cumulative regret we compare `Sampling-UCB` (with $\gamma = 0.5$) with the QRM1 algorithm by [16] and SR algorithm by [44]. We arbitrarily[4] choose the following reservoir: the arms are distributed according to a Bernoulli distribution with possible means $[0.5, 0.8]$ sampled with probabilities $[0.8, 0.2]$. We remark that the SR algorithm and `Sampling-UCB` are very similar, they both sample approximately $\log(T)/p^\star$ arms and run a regret minimizer algorithm on this set of arms. The only difference is that the SR algorithm relies on the MOSS algorithm. Whereas the QRM1 algorithm proceeds by progressively adding new arms. In particular this algorithm is anytime. In Figure 1 we compare the cumulative regret of the different algorithms for a fixed horizon $T = 20000$. We observe that `Sampling-UCB` behaves similarly to SR and that QRM1 performs slightly worst (maybe because of the adaptation to $T$). We also check that all algorithms have a regret that is logarithmic with the horizon as expected. To this aim, in Figure 2, we plot the cumulative regret (for the same reservoir) for all horizons $T \in \{100, 200, \ldots, 10000\}$.

**Best-arm identification**    For best arm identification we compare our algorithm with the BUCB algorithm by [32]. In Figure 3 we compare the performance of the algorithms across varying $\Delta$ for a fixed $T = 1000$. That is, we consider reservoirs of the form $[0.2, \Delta, 1]$ for $\Delta \in (0.01 \times i)_{i \in [79]}$ with probabilities $[0.29, 0.69, 0.02]$. The BUCB algorithm presents an issue as it is designed for the fixed confidence regime the algorithm takes $\delta$ as a parameter. We set $\delta$ equal to an arbitrarily low constant. The BUCB algorithm works by opening successively large brackets of arms, however as they do not provide results in high probability, only in expectation, they can draw significantly less arms from the reservoir. The performance of `Elimination` seems favourable compared to BUCB, however, one may be able to improve the performance of BUCB with parameter tuning.

---

[4]Which is not very important, since we evaluate the algorithms from a problem-dependent point of view

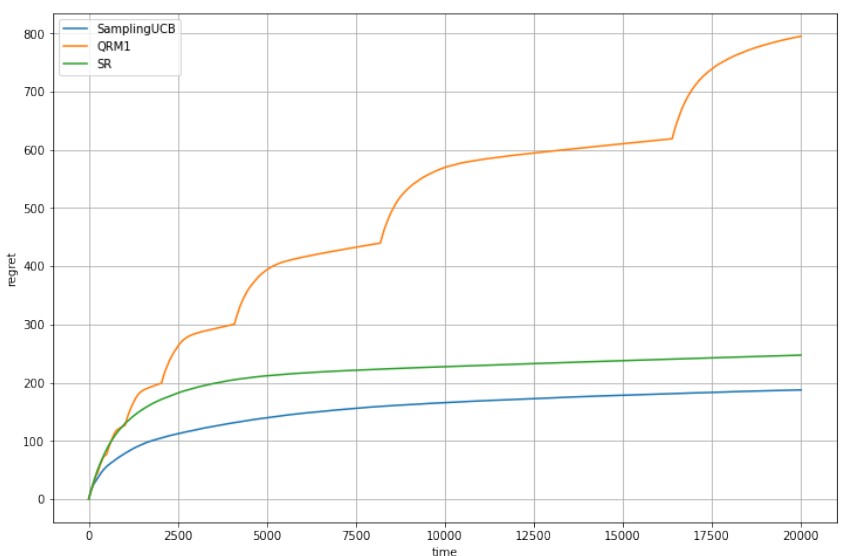

Figure 1: Cumulative regret in function of the time estimated by 100 Monte-Carlo simulations.

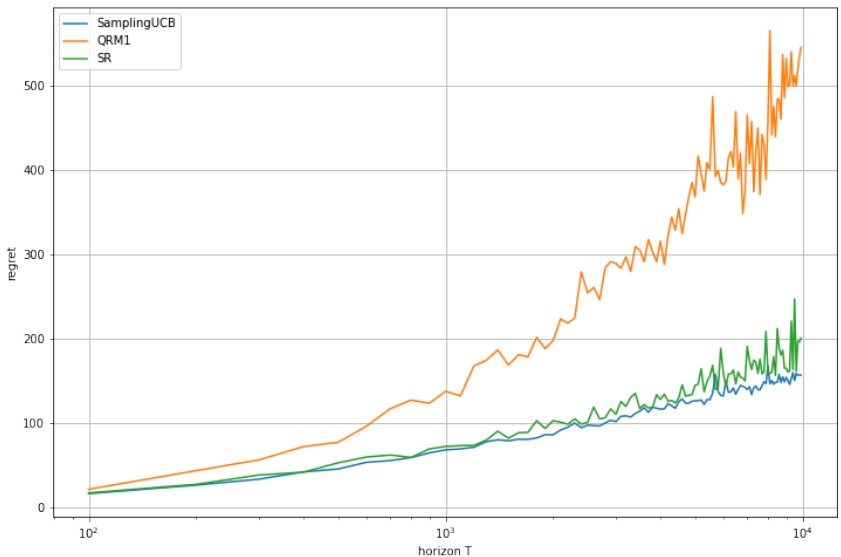

Figure 2: Cumulative regret in function of the horizon $T \in \{100, 200, \dots, 10000\}$ estimated by 100 Monte-Carlo simulations.

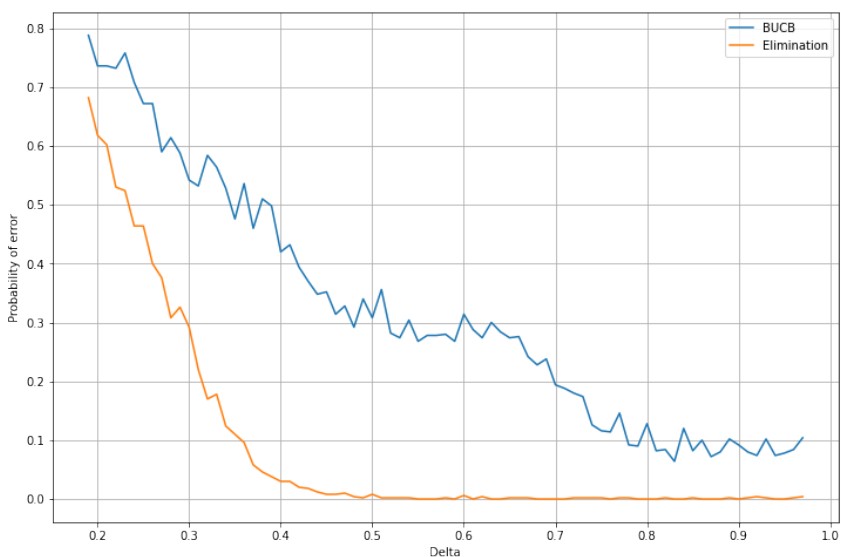

Figure 3: Probability of error for best arm identification across varying $\Delta$ using $500$ Monte-Carlo simulations.