# OpenReview forum: "Bandits with many optimal arms"
_NeurIPS.cc/2021/Conference — NeurIPS 2021 Poster_

### Official Review · Reviewer_C1hU · 2021-07-13

**Rating:** 6
**Confidence:** 3

**Summary:**

Authors consider the stochastic bandit setting considering a possibly infinite number of arms. They call p* the fraction of optimal arms and investigate both the cumulative regret minimization and the best-arm identification problems. For both problems they provide a lower bound and a solution matching the upper bound (up to log factors).

**Limitations And Societal Impact:**

See above.

**Main Review:**

Authors consider the stochastic bandit setting considering a possibly infinite number of arms. They call p* the fraction of optimal arms and investigate both the cumulative regret minimization and the best-arm identification problems.

If compared to existing works, they consider a more general setting and the obtained results are stronger than the ones obtained in the existing simpler frameworks.

For the first problem they provide a lower bound argument and an algorithm matching the bound up to multiplicative log factor of order 1/\Delta. The algorithm requires the knowledge of p* as input parameter, this seems to be a limitation, however authors shown that knowing such a parameter is strongly necessary.

Similarly, for the best-arm identification setting they provide a lower bound and an upper bound to the probability of choosing a suboptimal arm. The upper-bound matches the lower one up to a log factor in the exponential. The proposed algorithm does not require the knowedge of p* as it was for the regret-minimization solution.

I would like to know which challenges/technical difficulties have been solved in the proofs if compared to existing works.
Secondly, I wonder if there are connections with the combinatorial bandit problem.

**Time Spent Reviewing:**

5

---

> ### Author Response · Authors · 2021-08-10
> **Response**
>
> Technical difficulties compared to existing works
> For cumulative regret the main challenge is that one needs to sample approx $\log (T)/p^*$ arms in order to find at least one optimal arm with large enough probability. If a classical bandit algorithm is then applied to this set of arms, we would therefore have a regret of order $\log(T)^{2} / (p^* \Delta)$ - see the introduction where we discuss this. This happens to be sub-optimal, because in a $\log(T)/p^*$ sub-sample of arms, there are in fact with high enough probability $\log T$ (near) optimal arms - and this high proportion of optimal arms can be leveraged to obtain a regret of order $\log(T)/(p^* \Delta)$ instead of $\log(T)^2/(p^* \Delta)$. The intuition behind this phenomenon is the  following. First, having a high proportion of near-optimal arms improves the accuracy of estimation of the (near) optimal mean - and this phenomenon is not present in classical bandit theory where there is just one optimal arm. Second, in the $K$ armed bandit setting, with minimal gap $\Delta$, there are algorithms that achieve a constant regret of order $K/\Delta$ (up to $\log \Delta^{-1}$ terms) if the (near) optimal mean is known. So that, identifying $K$ with the size $\log(T)/p^*$ of the sub-sample of arms, we obtain the $\log (T)/(p^* \Delta)$ regret (up to $\log \Delta^{-1}$). Our algorithm, and our proof, do not exactly follow this scheme, but are close to it. We provide a more aggressive UCB that does not depend upon T, see Section 2.1 for details. This UCB works because of the phenomenon discussed before, namely that the optimal mean concentrates much faster in our setting. This is present in the proof when we control said optimal empirical mean. This is the main difference with the classical finite armed bandits problem.
>
> For best arm identification we must be adaptive on p* and cannot begin by choosing a well size sample from the reservoir. Therefore we begin with a very large set of arms, order $T$, and successively eliminate by a constant factor. Now, until our set is sufficiently small such that arms are pulled more than $\Delta^{-2}$ times we have no guarantees on the type of arms we remove. Thus, we need to ensure that when $t_i > \Delta^{-2}$ there is still a proportion of optimal arms of the order $p^*$. A more classical elimination algorithm would not have this guarantee.  We solve the problem by adding fresh arms to our set at each iteration. Our problem is made more difficult as we work in the fixed budget setting and must ensure we have a high proportion of optimal arms with very high (exponential in $T$) probability, as opposed to expectation. This is in contrast with the fixed confidence setting where more conservative algorithms can be used as one aims to bound stopping time only in expectation, which is much less restrictive. For instance in [28] they focus on the fixed confidence setting and their results do not hold in high probability, which would be necessary to compare with our own, see Remark 4 therein and the discussion in section 1.3 of our paper.
> Combinatorial bandits
>
> We do not see any particular links between our setting and the combinatorial bandits problem. In the combinatorial bandit problem, instead of one of K arms, a subset of arms $S \in 2^[K]$ is sampled and the rewards of all arms in $S$ are revealed. For example in the combinatorial pure exploration problem (e.g. used to find spanning trees, to do matching, etc.), the goal is to find an optimal set of arms $M^*$, whereas in our paper we are only concerned with finding one optimal arm - a much easier problem from an information theoretical perspective.

---

### Official Review · Reviewer_wUpn · 2021-07-16

**Rating:** 6
**Confidence:** 3

**Summary:**

This paper considers infinitely many armed bandit problems where the significant portion is optimal. In particular, p* fraction of the arms are optimal.
This paper considered two objectives (total reward optimization and best arm identification). For the first objective, the paper proposes a version of UCB algorithm with a slightly smaller confidence bound (sampling UCB, alg 1) that matches lower bound up to O(log(1/Delta)), whereas for the second objective the paper proposes a version of elimination (Elimination, alg 2) that matches lower bound up to O(log Delta^{-2}).

This paper is mainly theoretical and the assumption of many optimal arms is more restrictive than the existing literature of many-armed bandit problem (without empirical justification). I think the algorithms proposed in this paper is novel and the paper is marginally above the acceptance threshold as it is of theoretical interest of the readers. Having derived corresponding lower bounds based on change-of-measure discussion is also nice.
There can be some space for improvement on the structure of the paper. It spends 5+2/3 pages before the main results. As a result, results of the simulation are not presented during the main paper.

After the discussion period:
I would not change my rating strongly (will keep WA). I think the paper is above the threshold.



**Main Review:**

* Major comments:

** Total reward maximization

Sampling UCB (Algorithm 1) first draws C log T/p* arms and apply a version of UCB algorithm with slightly smaller confidence bound (sqrt(log Ni/Ni)). Such a confidence bound sometimes yields UCB value of U_i(t) < \mu_i. However, given there are p* fraction of arms, there is at least one optimal arm with U_i(t) >= \mu_i. This trick reduces an O(log T) factor compared with standard UCB.
- Can you briefly discuss where the log(1/Delta) gap between upper and lower bounds? The lower bound looks reasonable given it took log T/Delta**2 to identify the optimality of arm and 1/p rate of finding the optimal arm.

** Best-arm identification
Elimination (Algorithm 2) first draws T/2 arms and then at each epoch it discards the lower half of them and adds some new arms of the same order. The algorithm is aimed to draw some optimal arm tilde(O)(pDelta^2T) times and adopts a polynomially decreasing size of epochs. The algorithm is different from successive rejection and is interesting.

** Comparison with other algorithms
Can you compare the results here with similar work? For example, what kind of bounds can we obtain when we apply [39]?

** Simulation results:
It would be great if you briefly discuss the other algorithms in simulations.

* Minor comments:

- The paper may rename algorithm 2 as it not only eliminates some of the arms but also adds new ones at each epoch.
- Can you motivate the problem setting (a many-armed bandit with p* fraction of optimal arms)?
- Some of the equations referred in the main papers only appear in the supplemenatry materials. For example, Eq. (3).

**Time Spent Reviewing:**

6

---

> ### Author Response · Authors · 2021-08-10
> **Response**
>
> Thank you for  your detailed response and the time taken to read our paper. We will expand our experimental section to include a description of the QRm1 and SR algorithms.
>
> Equivalent settings - epsilon good arm, arm in a given quantile
>
> As mentioned in the abstract of our paper and paragraph “equivalent setting” page 2, our results apply directly to several equivalent settings - finding an epsilon good arm, finding an arm above a given quantile and competing against the jth best arm. For finding an $\epsilon$ good arm let $\mathcal{A}_\epsilon$ be the set of epsilon good arms and let $p^*_\epsilon$ be the proportion of epsilon good arms in the reservoir - that is an arm belongs to $\mathcal{A}_\epsilon$ with probability $p^*_\epsilon$. Define
>
> $$\mu_\epsilon^- := \min(\mu_a :a \in \mathcal{A}_\epsilon)\;,$$
>
> as the smallest epsilon good arm and let $\Delta_\epsilon =  \mu_\epsilon^- - \max(\mu_a : a \in \mathcal{A}/\mathcal{A}_\epsilon)$
>
> For Algorithm 1, if we instead take $L$ of the order $\log(T)/p^*_\epsilon \gamma^2$  we will ensure we have a high proportion of epsilon good arms in our initial set with high probability. Then to adapt the proof one can look at the probability of the UCB bounds exceeding  $\mu_\epsilon^-$, as opposed to $\mu^*$. Following this analysis one sees we recover the same bound with $\Delta_\epsilon $ and $p^*_\epsilon$ replacing $\Delta$ and $p^*$ respectively.
>
> For Algorithm 2 we can adapt the proof by redefining $M_i$ as,
>
>  $$M_i = \big|\{a\in \mathcal A_i \cap \mathcal{A}_\epsilon:\hat \mu_i(a) \geq \mu_\epsilon^- -  \Delta_\epsilon/2\}\big|\,,$$
>
> and $N_i$ as,
>
> $$N_i = \big|\{a \in \mathcal A_i \cap  \mathcal{A}/\mathcal{A}_\epsilon:\hat \mu_i(a) \geq \mu^* - \Delta/2\}\big|\,.$$
>
> All bounds on $q_i$ should then hold (replacing $\Delta$ with $\Delta_\epsilon$, and the proof follows.
>
> In the case of identifying an arm above a given quantile $p^*$ would then be the quantile and $\Delta$
>
> In the case of finding an arm above a arm’s given quantile, $p^*$ is simply identical to the quantile, and the gap $\Delta$ is the difference between the mean of the $p^*/2$ best arm, and the mean of the $p^*$ best arm. To adapt our results we follow similar to the above. Indeed, in our setting our results improve on those of [4] and [28] which consider the problem of finding an epsilon good arm and quantile estimation respectively, see section 1.3.
> Thus, we are not restricted to a setting where a proportion of arms are exactly optimal. This allows our results to be much more adaptive in terms of practical application. Furthermore we allow for an infinite reservoir and our bounds do not scale with the total number of arms, rather the proportion $p^*$ of optimal arms. This should also lend our results to practical applications.  Our results also extend to finite bandits and we match the classical UB and LB up to log terms.
>
>
> Comparison of our results with [39], see section 1.3 for others
>
> When comparing with [39] one must keep in mind they work in the problem independent setting, which is a very different regime, and our bounds cannot be deduced from theirs. Their bounds do not depend upon the gap $\Delta$, they recover a bound of the order $\log(T)^{5/2}sqrt{\frac{T}{p^*}}$. They are also not optimal in the $\log$ and are not truly adaptive on the proportion of optimal arms as their algorithm requires a user defined constant that has a semi dependence on $p^*$ to be optimal. For further discussion on the comparison of our results to others see section 1.3 where we discuss other papers in the problem independent setting. Regardless, this is an important reference which we missed and we will include it.
>
> log(\Delta) gap between UB and LB in cumulative regret setting
>
> In the classical bandit setting one typically sets the UCB dependent on $T$ such that it is exceeded with probability less than $1/T$. In our setting we can exploit the fact there is a high proportion of optimal arms to have a much looser UCB not dependent upon  $T$ that is exceeded less than some constant probability. However as we do not know the value of $\mu^*$ or $\Delta$ we must ensure at least one optimal arm has a UCB above $\mu^*$ at each time step, this gives rise to the $\log(n)$ term in the UCB, see line 526 in the proof of Theorem 1, which in turn causes the additional $\log(1/\Delta)$ term in the final result. Essentially this term can be seen as the cost we pay for estimating $\Delta$.

---

### Official Review · Reviewer_Y2aD · 2021-07-17

**Rating:** 8
**Confidence:** 4

**Summary:**

The paper studies multi-armed bandits when there are a large number of arms, but when a constant
fraction of the arms are optimal. The paper studies methods for cumulative regret in the fixed
horizon setting and best arm identfication (BAI) in the fixed budget setting. In the former, they
describe an UCB-style algorithm which knows this constant fraction (p*) and bound its regret; they
complement this with a lower bound and an impossibility result which states that p* is necessary. In
the latter, the describe a successsive halving style algorithm and show that it nearly matches a
lower bound.

While I do have some reservations about the set up and the motivation, my overall perspective is
quite positive. The main results (theorems 1, 2, 4, 5) build on known techniques, but there is
also a reasonable amount of new ideas. While I did not have time to check the proofs, the proof
intuitions make sense and the results are believable. The paper is also well-written.


**Main Review:**

- In my opinion, the main shortcoming of the paper is that the setting lacks practical motivation.
  This, in and of itself is not a problem (e.g. K-armed bandits are rarely used in their simplest
  form in practice, but the intuitions and proof ideas have been used to study richer feedback
  models, such as linear, X-armed, and GP bandits which are used in practice). Unfortunately, I
  don't see a natural extension of this setting to richer feedback models. It would be helpful, if
  the authors can discuss how their setting and ideas can be enhanced in future work which might
  consider more practical settings, either using such richer feedback models or otherwise.
    - The motivating example the authors use in lines 30-33 is that there might be many arms in
      image classification, personalized medicine, or hyperparameter tuning. But the arms in these
      applications can be featurized, and the usual approach for tackling such a large number of
      arms is to use a richer feedback model so that the regret bounds scale with the complexity
      of the model and not the number of arms. It is also unreasonable to assume that a constant
      fraction of arms in these applications will be exactly optimal (usually there will be several
      good arms, but their means may not be the same), let alone that p* will be known ahead
      of time.

- Please comment on extending Algorithm 1 to the any-time setting? It appears that sampling of the
  initial pool can be handled by slowly admitting more arms after several rounds, but I am not sure
  if the confidence intervals would still work.

- The UCB used by the authors in (2) is reminiscent of the UCB used in the following paper and is
  probably worth citing.
    Jamieson et al, "lil’ UCB : An Optimal Exploration Algorithm for Multi-Armed Bandits"

- Can you explain the proof intuition in Theorem 3?

- Have you explored other conditions, besides the constant p* fraction, under which Algorithm 2 (or
  any other sensible algorithm) will return a ``good'' arm?
    - For instance, there are several arms, but you only need to find an \epsilon optimal arm or
      an arm whose mean is in some top percentile.

- The successive halving strategy that the authors have used for the BAI setting was first
  introduced by Karnin et al, "Almost Optimal Exploration in Multi-Armed Bandits" and should be
  cited.


-------------------------------

Post-rebuttal: Based on the authors' reply, I have increased my score to 8. While I am less convinced about the practical motivation given by the authors and the results for the cumulative regret problem, I think the results for BAI are potentially interesting and the abstraction could be useful for some follow up work. While the proofs fundamentally build on well-known techniques, they also use some new ideas.

**Time Spent Reviewing:**

3

---

> ### Author Response · Authors · 2021-08-10
> **Response**
>
> Thank you for  your detailed response and the time taken to read our paper. Also for the mentioned references, which we will include.
>
> Practical applications
>
> Here are several possible applications of our setting:
> -Automated hiring of crowd workers (most biased coins problem): see [24] and reference therein.
> -Hyperparameter optimization: the reservoir is the set of hyperparameter, the user fix a quantile p^* and the goal is to find a hyperparameter which is in the best p^* fraction of the reservoir, see Li et al., Hyperband: A Novel Bandit-Based Approach to Hyperparameter Optimization (2018).
> See also the applications mentioned by Berry et al., Bandit problems with infinitely many arms (1997). We agree that in general if some side information or prior knowledge on the structure of the arms is available one should take advantage of it.  We rather consider in this paper the agnostic case with no extra information also encountered in practice, e.g. hyperparameter optimization.
>
> Equivalent settings - epsilon good arm, arm in a given quantile
>
> As mentioned in the abstract of our paper and paragraph “equivalent setting”, Page 2, our results apply directly to several equivalent settings, including finding an epsilon best arm and finding an arm in a given quantile or “top percentile”
>
> For finding an $\epsilon$ good arm let $\mathcal{A}_\epsilon$ be the set of epsilon good arms and let $p^*_\epsilon$ be the proportion of epsilon good arms in the reservoir - that is an arm belongs to $\mathcal{A}_\epsilon$ with probability $p^*_\epsilon$. Define
> $$\mu_\epsilon^- := \min(\mu_a :a \in \mathcal{A}_\epsilon)$$
> As the smallest epsilon good arm and let $\Delta_\epsilon =  \mu_\epsilon^- - \max(\mu_a : a \in \mathcal{A}/\mathcal{A}_\epsilon)$
>
> For Algorithm 1, if we instead take $L$ of the order $\log(T)/p^*_\epsilon \gamma^2$  we will ensure we have a high proportion of epsilon good arms in our initial set with high probability. Then to adapt the proof one can look at the probability of the UCB bounds exceeding  $\mu_\epsilon^-$, as opposed to $\mu^*$. Following this analysis one sees we recover the same bound with $\Delta_\epsilon $ and $p^*_\epsilon$ replacing $\Delta$ and $p^*$ respectively.
>
> For Algorithm 2 we can adapt the proof by redefining $M_i$ as,
>
>  $$M_i = \big|\{a\in \mathcal A_i \cap \mathcal{A}_\epsilon:\hat \mu_i(a) \geq \mu_\epsilon^- -  \Delta_\epsilon/2\}\big|\,,$$
>
> and $N_i$ as,
>
> $$N_i = \big|\{a \in \mathcal A_i \cap  \mathcal{A}/\mathcal{A}_\epsilon:\hat \mu_i(a) \geq \mu^* - \Delta/2\}\big|\,.$$
>
> All bounds on $q_i$ should then hold (replacing $\Delta$ with $\Delta_\epsilon$), and the proof follows.
>
> In the case of identifying an arm above a given quantile $p^*$ would then be the quantile and $\Delta$
>
> In the case of finding an arm above a arm’s given quantile, $p^*$ is simply identical to the quantile, and the gap $\Delta$ is the difference between the mean of the $p^*/2$ best arm, and the mean of the $p^*$ best arm. To adapt our results we follow similar to the above. Indeed, in our setting our results improve on those of [4] and [28] which consider the problem of finding an epsilon good arm and quantile estimation respectively, see section 1.3.
> Thus, we are not restricted to a setting where a proportion of arms are exactly optimal. This allows our results to be much more adaptive in terms of practical applications. Furthermore we allow for an infinite reservoir and our bounds do not scale with the total number of arms, rather the proportion $p^*$ of optimal arms. This should also lend our results to practical application.
> Intuition behind proof of Theorem 3
> The intuition behind the proof of theorem 3 is to consider two reservoir distributions, the first has two arms B(1/2), B(1/2 - \Delta) with proportions p* and 1-p* respectively. The second has three arms B(1/2+ \Delta) ,B(1/2), B(1/2 - \Delta) with proportions q*, p* and 1-p* - q* respectively. With a coupling argument one can relate the KL divergence between the sampling distributions on the two reservoirs directly to the number of times arms not equal to B(1/2) are pulled. If the regret is low on the first reservoir distribution, arms following B(1/2) must be pulled many times, hence the divergence must also be low.
> The second step is to consider an event where one samples an arm with mean 1/2 many times corresponds to a low regret on the first reservoir but a high regret on the second. Roughly speaking if this event occurs with high probability on the first reservoir it must also on the second, as the divergence is small, and the learner suffers a high regret.
> Algorithm 1 anytime
> To make Algorithm 1 anytime one should be able to grow the number of arms slowly, keeping a subsample size of order $\log(t)/p^*$, where $t$ is the current time. We believe this should work without much additional work.
>
> Mentioned references
>
> Indeed a successive halving strategy for BAI is used by Karnin et al, "Almost Optimal Exploration in Multi-Armed Bandits" although without the trick of adding fresh arms as they don’t need to be adaptive to $p^*$.  Also in Jamieson et al, "lil’ UCB : An Optimal Exploration Algorithm for Multi-Armed Bandits" they consider an algorithm which, like our own uses a UCB which does not depend on the time horizon $T$ but only the number of times an arm has been pulled. Although they do so for different reasons, namely to adapt to the infinite time horizon of the fixed confidence setting.
>
> Regardless, both are clearly important references. We thank the reviewer for highlighting them and will add citations for both.

---

### Decision · Program_Chairs · 2021-09-27

**Decision:**

Accept (Poster)

**Comment:**

All reviewers are positive about this paper. The reviewers agree that the paper shows interesting new ideas in the results (especially ones for best arm identification). The reviewers also raise concerns about practical motivations and the presentation structure of the paper. I think the paper can benefit from improving in these directions.